# Targeting the latent human cytomegalovirus reservoir for T-cell-mediated killing with virus-specific nanobodies

Timo W. M. De Groof [1,2,6], Elizabeth G. Elder [3,5,6], Eleanor Y. Lim[3], Raimond Heukers [1,4], Nick D. Bergkamp[1], Ian J. Groves [3], Mark Wills[3], John H. Sinclair[3,7] & Martine J. Smit [1,7 ✉]

Latent human cytomegalovirus (HCMV) infection is characterized by limited gene expression, making latent HCMV infections refractory to current treatments targeting viral replication. However, reactivation of latent HCMV in immunosuppressed solid organ and stem cell transplant patients often results in morbidity. Here, we report the killing of latently infected cells via a virus-specific nanobody (VUN100bv) that partially inhibits signaling of the viral receptor US28. VUN100bv reactivates immediate early gene expression in latently infected cells without inducing virus production. This allows recognition and killing of latently infected monocytes by autologous cytotoxic T lymphocytes from HCMV-seropositive individuals, which could serve as a therapy to reduce the HCMV latent reservoir of transplant patients.

[1] Amsterdam Institute for Molecular and Life Sciences (AIMMS), Division of Medicinal Chemistry, Faculty of Science, VU University, De Boelelaan 1108, Amsterdam, The Netherlands. [2] Department of Medical Imaging, In Vivo Cellular and Molecular Imaging Laboratory (ICMI), Vrije Universiteit Brussel (VUB), Laarbeeklaan 103, Brussels, Belgium. [3] Department of Medicine, Addenbrooke's Hospital, University of Cambridge, Cambridge, United Kingdom. [4] QVQ Holding BV, Yalelaan 1, Utrecht, The Netherlands. [5] Present address: Public Health Agency of Sweden, Nobels väg 18, Solna, Sweden. [6] These authors contributed equally: Timo W. M. De Groof, Elizabeth G. Elder. [7] These authors jointly supervised this work: John H. Sinclair, Martine J. Smit. ✉email: mj.smit@vu.nl

Latent reservoirs of viral pathogens are significant barriers to the eradication of these viruses from their hosts[1–3]. During latency, human herpesviruses and retroviruses maintain their viral genomes in the absence of infectious virus particle production, often with limited viral gene expression[4]. As such, latent infections are refractory to treatment with typical antivirals that target replication of the virus[2]. Furthermore, the low number of latently infected cells and the relatively low levels of viral gene expression during latency reduces the levels of viral antigens that would otherwise be readily detectable by the host immune system[3]. Reactivation from latency results in the dissemination and reseeding of the virus. In the case of the ubiquitous beta-herpesvirus human cytomegalovirus (HCMV), such sporadic reactivation events are well controlled by a combination of cellular and humoral immunity[3]. In particular, cytotoxic T lymphocytes (CTLs) against the HCMV immediate-early (IE) antigens are present at high frequency in HCMV-seropositive individuals[5]. In immunocompromised or immunosuppressed individuals, this control of reactivation is lost, and for both solid organ and stem cell transplant patients, HCMV reactivation frequently results in a disseminated viral infection that is a major cause of transplant rejection and mortality[3].

Targeting the latent viral reservoir in graft donors and recipients could lower the incidence and severity of HCMV-associated disease in transplant patients[3]. Latently infected CD34$^+$ hematopoietic progenitor cells and their derived CD14$^+$ monocytes suppress IE gene expression via the remodeling of chromatin structure and multiple repressive transcription factors at the viral major immediate-early promoter/enhancer region (MIEP)[6]. By using pan-specific histone deacetylase (HDAC) inhibitors, we previously showed that transient activation of lytic IE gene expression in latently infected monocytes results in CTL-mediated killing of infected cells[7]. However, we wished to develop a virus-specific molecule that would limit off-target effects but, similarly, induce IE gene expression for use as a shock-and-kill therapeutic.

The viral protein US28, a chemokine receptor with high homology to human chemokine receptors, is expressed during HCMV latency[8–10]. Importantly, a number of reports using particular models of HCMV latency have shown that US28 signaling is essential for the establishment and maintenance of HCMV latency, which is due, at least in part, to US28-mediated repression of the major IE promoter[8,11–13]. Previously, a monovalent, antagonistic nanobody targeting US28 was generated[14].

Here, we show the partial reactivation and CTL-mediated killing of latently infected cells via a bivalent nanobody, VUN100bv, derived from this previously described monovalent nanobody. VUN100bv binds and partially inhibits signaling of the viral protein US28. We show that partial inverse agonistic activity of VUN100bv results in transient activation of the MIEP and subsequent IE gene expression during latency, but no full virus reactivation. Consequently, VUN100bv treatment drives recognition and killing of latently infected monocytes by CTLs, and demonstrates the efficacy of VUN100bv for lowering latent viral loads in ex vivo experimentally infected peripheral blood mononuclear cells (PBMCs).

## Results

### Generation of a partial inverse agonistic US28-targeting nanobody.
The virally encoded chemokine receptor US28 is absolutely essential for HCMV latency in a number of myeloid cell models of HCMV latency[8,11,12]. Consistent with this, inhibition of US28 using the small-molecule US28 inhibitor VUF2274 results in untimely reactivation of the full viral lytic transcription program and production of new infectious viral particles[12]. However, VUF2274 also shows substantial toxicity and is known to inhibit general CCR1 signaling[12,15]. Consequently, we reasoned that new, highly specific reagents that target and inhibit US28 would be needed for any safe shock-and-kill strategy.

To this end, we developed a new partial inverse agonistic US28-targeting nanobody in the hope that it would inhibit US28 function and efficiently induce IE gene expression for subsequent targeting by host IE-specific CTLs. Nanobodies targeting the extracellular domains of several chemokine receptors are antagonistic as monovalent formats and display inverse agonistic properties as bivalent nanobodies[16,17]. We, therefore, developed a bivalent format of our existing nanobody (VUN100), which we termed VUN100bv[14]. The monovalent nanobody VUN100 displaces US28 endogenous ligands and binds the extracellular domains of US28 with high affinity[14]. VUN100bv was created by fusing two VUN100 molecules using a 30GS linker. This fusion of two VUN100 molecules showed an approximately 10-fold increase in binding affinity compared to the monovalent VUN100 (0.2 nM ± 0.1 vs 2 nM ± 1) (Fig. 1a). Similarly, VUN100bv displaced $^{125}$I-labeled CX3CL1, a known US28 ligand, with approximately 10-fold higher pKi compared to monovalent VUN100 (9.4 ± 0.4 vs 8.1 ± 0.1) (Fig. 1b). Next, we tested the functional effect of VUN100bv on NFAT (Nuclear Factor of Activated T cells) signaling, which is induced by US28 wildtype (WT) receptor and several US28 mutants in HEK293T cells (Fig. 1c)[18]. These mutants included US28 ΔN (2–22) mutant (lacking the entire N-terminus, unable to bind US28 ligands but signals in a constitutive manner), US28 Y16F mutant (unable to bind chemokines but signals in a constitutive manner), and a US28 R129A mutant (US28R$^{3.50}$A, unable to couple to G proteins and signal in a constitutive manner). To determine the functional effects of the US28 nanobodies on US28 activity, HEK293T cells were co-transfected with a vector expressing HA-tagged US28 (mutants) and a vector containing the luciferase gene under the control of an NFAT-promoter. We confirmed that the expression levels of the different US28 constructs were similar to ensure that the observed differences in US28-mediated NFAT activity are solely due to differences induced by the US28 nanobodies (Supplementary Fig. 1). Consistent with previous studies, we found that US28-mediated NFAT activation was dependent on constitutive signaling as NFAT activation was observed for all different US28 constructs except for the US28 R129A G protein uncoupled mutant[18]. Then, we analyzed the effect of VUN100bv on US28 constitutive activity. When added to cells expressing US28 WT or US28 Y16F mutant receptor, VUN100bv inhibited US28 constitutive activity by 44.9 (±0.9)% and 46.7 (±0.6)%, respectively. As predicted, this inhibitory effect of VUN100bv was not seen for the US28 ΔN (2–22) mutant, since binding of VUN100(bv) to US28 depends on the N-terminus[14]. No significant effects on US28-mediated NFAT activity were observed for the monovalent VUN100 or a non-targeting nanobody. We also confirmed that a vast excess of the monovalent VUN100 nanobody could not recapitulate the inverse agonist properties of VUN100bv. Even at a 10-fold higher concentration (1 μM) of monovalent VUN100, corresponding to 400-fold times its $K_d$ value, no inhibition of constitutive signaling of US28 to NFAT was observed (Supplementary Fig. 2). This strongly indicates that VUN100 has no inverse agonistic activity. Taken together, our results show that VUN100bv acts as an antagonist by competing for binding of CX3CL1 to US28 and as a partial inverse agonist by inhibiting the constitutive activity of US28.

Next, we evaluated the binding to US28 and inverse agonistic activity of VUN100bv in the monocytic THP-1 cell line, an established model for HCMV latency[19]. Both VUN100 and

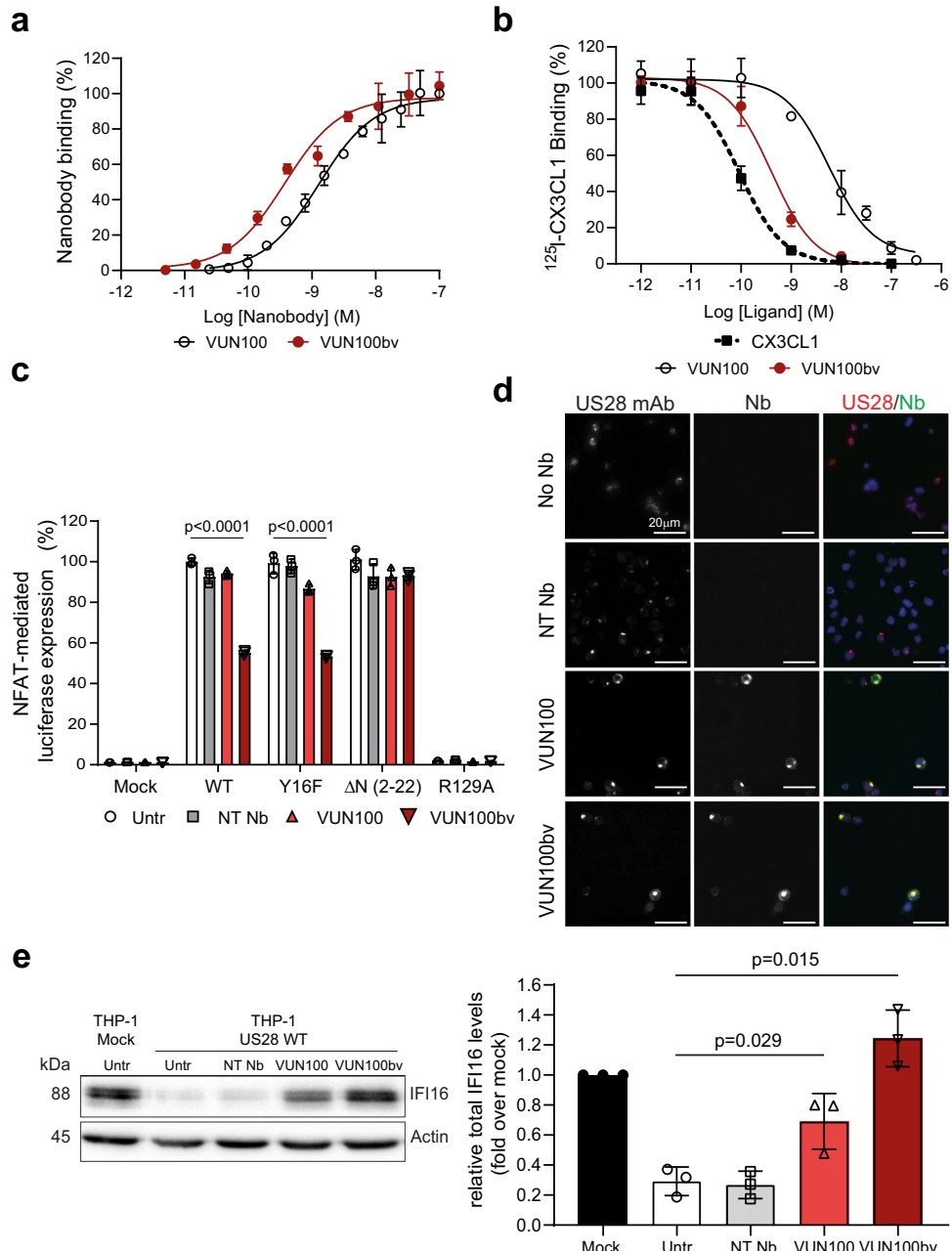

**Fig. 1 VUN100bv binds and inhibits US28 signaling. a** ELISA binding of monovalent VUN100 and bivalent VUN100bv to membrane extracts of US28-expressing HEK293T cells. Representative figure of three independent experiments. **b** Displacement of $^{125}$I-CX3CL1 from US28-expressing membranes by unlabeled ligand or the nanobodies VUN100 and VUN100bv. Representative figure of three independent experiments. **c** Effect of nanobodies on US28-mediated NFAT (nuclear factor of activated T cells) activation. HEK293T cells expressing either NFAT-luciferase reporter only (Mock) or NFAT-luciferase reporter together with US28 wildtype receptor (WT), US28 Y16F mutant (Y16F), US28 ΔN (2–22) mutant (ΔN (2–22)) or US28 R129A mutant (R129A). Cells were untreated (untr) or treated with a non-targeting nanobody (NT Nb), VUN100, or VUN100bv for 24 h prior to luminescence measurement. Data were normalized to the untreated WT samples. Representative figure of three independent experiments. **d** Immunofluorescence microscopy of nanobody binding to US28-expressing THP-1 cells. US28 was detected using a polyclonal rabbit-anti-US28 antibody (US28 mAb). Cells were incubated without nanobody (No Nb), an NT Nb, VUN100, or VUN100bv. Bound nanobody was detected using the Myc-tag present on the nanobodies and an anti-Myc antibody (Nb). Representative figure of three independent experiments. **e** Western blot detection for total IFI16 levels of lysates of untreated THP-1 mock transduced cells (THP-1 Mock) or US28-expressing THP-1 cells (THP-1 US28 WT). THP-1 US28 WT cells were untreated (Untr) or treated with NT Nb, VUN100, or VUN100bv for 48 h. IFI16 protein levels were determined and normalized to actin protein levels. Relative IFI16 protein levels were normalized to untreated THP-1 mock cell lysates. $n = 3$ independent experiments from three independent biological replicates. All data are plotted as mean ± S.D. For all data, except for Fig. 1c, statistical analyses were performed using an unpaired two-tailed $t$ test. For Fig. 1c, statistical significance was determined using the Holm–Sidak method (two-sided with alpha = 0.05). Source data are provided as a Source Data file.

VUN100bv bound to US28-expressing THP-1 cells, unlike the non-targeting nanobody (Fig. 1d). In addition, none of the three nanobodies bound to mock transduced THP-1 cells, which do not express US28, indicating that these nanobodies are specific to our target (Supplementary Fig. 3). We then assessed the effect of the anti-US28 nanobodies on US28-mediated signaling in THP-1 cells by assessing IFI16 protein levels (Fig. 1e). IFI16 is downregulated by WT US28, but not the US28 R129A G protein uncoupled mutant, to support the repression of the MIEP[20]. VUN100bv treatment of US28-expressing THP-1 cells resulted in full restoration of total IFI16 protein levels while this was not seen for the non-targeting nanobody. Interestingly, VUN100 treatment also partially restored IFI16 protein levels. Altogether, our results show that, while both VUN100 and VUN100bv can bind to US28, only VUN100bv is able to consistently inhibit constitutive US28 signaling in both HEK293T cells and monocytic THP-1 cells.

**US28 nanobodies induce IE expression in infected CD14[+] monocytes.** Because repression of HCMV MIEP is a downstream consequence of US28 signaling in latently infected myeloid cells, we hypothesized that US28 inhibition by the inverse agonist VUN100bv might drive the inability to establish or maintain latency via the (re)activation of viral IE expression from the MIEP in otherwise latently infected cells. Consequently, we determined the effect of the US28 nanobodies on the establishment of latency in infected monocytes. Primary CD14[+] monocytes were isolated, infected with HCMV for 2 h, and treated afterward with nanobodies. At two and 6 days post infection, IE expression was assessed (Fig. 2a, b and Supplementary Fig. 4). As a positive control for induction of lytic viral gene expression in these assays, we treated monocytes with the phorbol ester PMA (phorbol myristate acetate), which induces differentiation of monocytes to a macrophage-like phenotype and is known to result in reactivation of HCMV lytic infection within 24–48h of treatment rather than the 5–7 days needed for induction of reactivation by differentiation of monocytes to monocyte-derived mature dendritic cells (mDCs) by GM-CSF/lipopolysaccharide (LPS)[13,21]. As expected, PMA treatment resulted in an increase in IE expression. VUN100bv treatment also resulted in an increase in IE-expressing monocytes compared with untreated or non-targeting nanobody-treated monocytes (Fig. 2a, b). Interestingly, we saw a small but significant increase in IE expression with the antagonistic monovalent VUN100 in three out of four donors (Fig. 2a, b, Supplementary Fig. 4 and Supplementary Table 1). To ensure no bias in quantifying IE-positive cells, IE expression at 2 and 6 days post infection was also quantified using an automated plate reader (Supplementary Fig. 5). Similar results were obtained using automated quantification validating the results obtained via manual counting. To quantify full viral reactivation and subsequent virus production, latently infected cells were co-cultured with indicator fibroblasts, a cell type permissive for lytic infection, after nanobody treatment. We then quantified the formation of IE2-eYFP-positive infectious foci, a consequence of viral infection of the indicator fibroblasts, to determine the level of virus production. Importantly, none of the nanobodies, including VUN100bv, resulted in any significant IE focus formation (Fig. 2c). In contrast, the production of infectious virus from latently infected monocytes was induced with PMA treatment.

To ensure that VUN100bv also reverses latency and not only prevents the establishment of latency mediated by US28, we performed similar experiments but added the nanobodies or PMA after the establishment of latency at 6 days post infection, a time frame that routinely establishes latent infection in primary monocytes. Six days post infection and before treatment, no

significant differences in IE expression between the different wells were observed (Supplementary Fig. 6). Again, VUN100bv treatment resulted in a significant upregulation of IE expression compared with the untreated or non-targeting nanobody-treated monocytes (Fig. 2d). Treatment with monovalent VUN100 also resulted in a very small but significant upregulation of IE expression. Importantly, no production of infectious viral particles was observed from the untreated or nanobody-treated monocytes upon co-culturing with fibroblasts for eight days (Fig. 2F). In contrast, co-culturing of PMA-treated latently infected monocytes with fibroblasts resulted in a significant upregulation of IE2-eYFP-positive infectious foci formation. Moreover, to ensure that the effect of VUN100bv is US28-specific, we performed the same experiments with Titan WT and Titan ΔUS28 virus (Supplementary Fig. 7). Also in this setting, using the Titan WT virus, we noticed a significant upregulation of IE expression upon VUN100bv or PMA treatment of latently infected cells compared to the untreated or non-targeting nanobody-treated cells. In contrast, using the Titan-ΔUS28 virus, increase in IE expression was observed, caused by the lack of US28 expression by this virus[12]. However, IE expression could be further increased by PMA treatment. Treatment of the Titan-ΔUS28-infected CD14+ monocytes with the non-targeting nanobody resulted in a small, but not significant, increase of IE expression compared to the untreated cells. However, most importantly, VUN100bv treatment did not result in higher IE expression compared with the non-targeting nanobody, indicating that the effect of VUN100bv is US28-specific. Taken together, these results indicate that VUN100bv treatment results in only a partial reactivation of the viral lytic transcription program in a US28-specific manner in latently infected CD14[+] monocytes. While reactivating the monocytes at the level of IE expression, the nanobody treatments do not result in reactivation of full virus production.

To analyze the extent of reactivated lytic gene expression induced by VUN100bv in more detail, we assessed viral gene expression in latently infected monocytes treated with nanobodies. To do this, we also used monocytes treated with PMA, which induces myeloid differentiation and permits full lytic infection. At 6 days post infection, RNA was isolated and gene expression of different markers was tested by RT-qPCR (Fig. 3). As expected, PMA treatment of infected monocytes resulted in increased levels of transcripts from the major IE *IE72* gene, the early *UL44* gene, the late *UL32* gene, and the *US11* immune evasion gene (Fig. 3). Moreover, we noticed that the addition of the non-targeting nanobody resulted in a small induction of gene expression of IE72, UL44, and US11, which could be owing to a non-specific effect of the non-targeting nanobody, residual contaminants, or simply manipulation of the cells.

Consistent with Fig. 2, VUN100bv treatment of infected CD14[+] monocytes resulted in increased levels of the major IE *IE72* transcript, compared to the non-targeting nanobody treatment (Fig. 3a). Although VUN100bv treatment resulted in a significant upregulation of major IE *IE72* transcript levels compared with the untreated samples, this was not the case for the non-targeting nanobody. In contrast, expression of the early *UL44* gene was only slightly increased by VUN100bv compared with non-targeting nanobody or VUN100 treatment, suggesting that viral DNA replication would not be optimal in these cells (Fig. 3b). Importantly, VUN100bv treatment did not result in *UL32* gene expression (Fig. 3c), which encodes the true late virion-associated protein pp150. This is consistent with our inability to detect infectious virus production in these cells (see Fig. 2).

Finally, we were minded that any shock-and-kill strategy could be thwarted by the expression of viral immune evasins. We therefore also assessed the effect of nanobody treatment on the

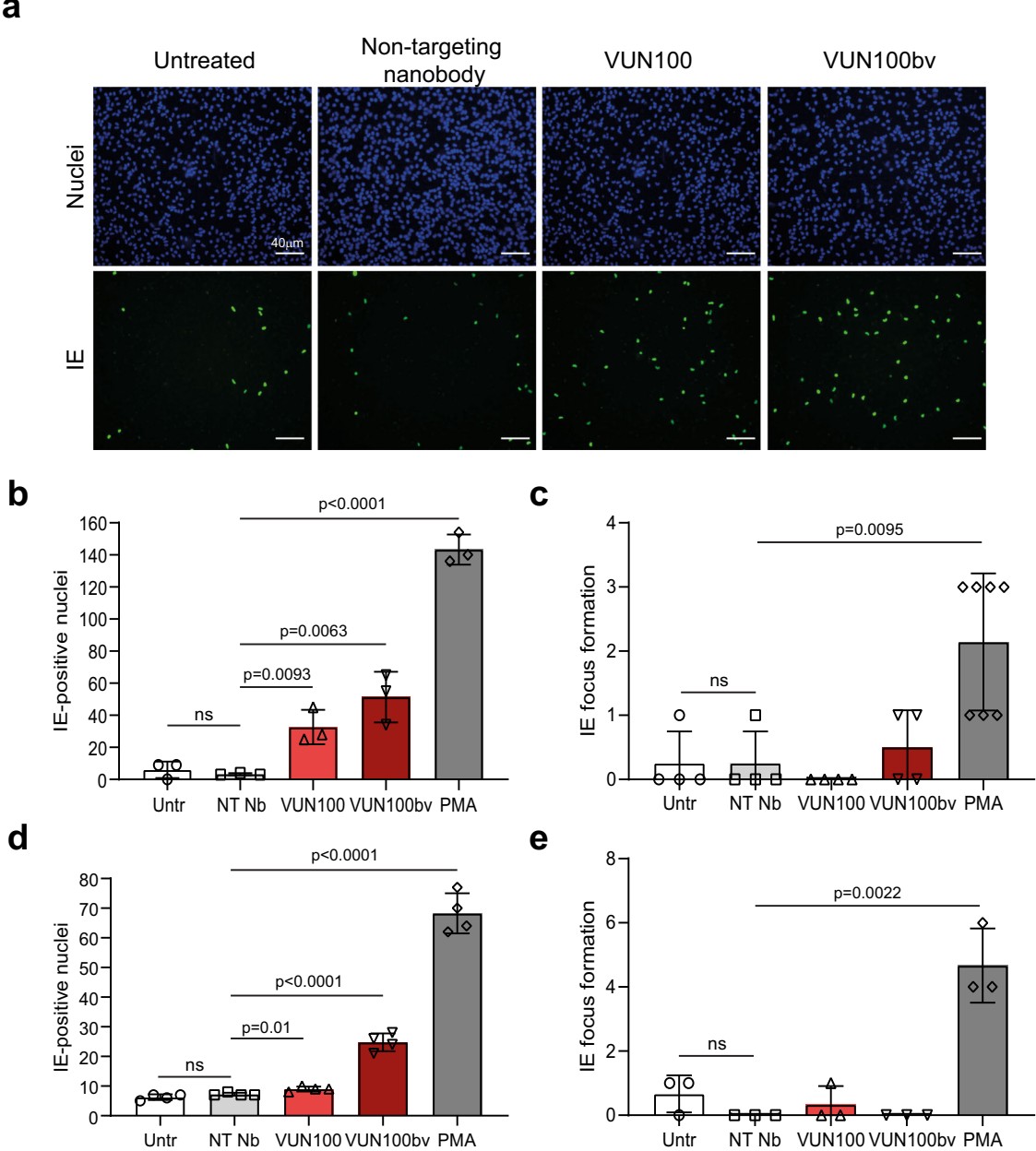

**Fig. 2 VUN100bv induces immediate-early expression but no full viral reactivation. a** CD14$^+$ monocytes were isolated, infected with HCMV IE2-eYFP, and were left untreated or treated with a non-targeting nanobody, VUN100 or VUN100bv. Two days post infection, cells were fixed and stained for immediate-early (IE) expression. **b** CD14$^+$ monocytes were isolated, infected with HCMV IE2-eYFP, and were left untreated (Untr) or treated with a non-targeting nanobody (NT Nb), VUN100, or VUN100bv. As a positive control, CD14$^+$ monocytes were pre-treated with 20 ng/ml PMA before infection (PMA). IE-positive nuclei were counted 6 days post infection. **c** Six days post infection, untreated (untr), nanobody-treated monocytes (NT Nb, VUN100, and VUN100bv) or monocytes pre-treated with 20 ng/ml PMA before infection (PMA) were co-cultured with Hff1 fibroblasts. IE-positive infectious foci formation was quantified after 4 days of co-culturing. **d** Six days post infection, HCMV-infected CD14+ cells were left untreated (Untr) or were treated with NT Nb, VUN100, VUN100bv, or 20 ng/ml PMA (PMA). IE-positive nuclei were counted 8 days post infection. **e** Eight days post infection, untreated (Untr), nanobody-treated (NT Nb, VUN100, and VUN100bv) or PMA-treated monocytes were co-cultured with Hff1 fibroblasts. IE-positive infectious foci were quantified after 8 days of co-culturing. Representative figures, showing technical replicates, from two (Fig. 2d, e) or four (Fig. 2a b) biological replicates are shown. All data are plotted as mean ± S.D. For all figures, statistical analyses were performed using unpaired two-tailed *t* test. ns, *p* > 0.05. Source data are provided as a Source Data file.

expression of the immune evasion gene *US11* (Fig. 3d). In contrast to PMA, the slight increase of *US11* gene expression was discernibly and significantly lower for the nanobodies compared to differentiation-induced reactivation. Taken together, these results confirm that VUN100bv treatment results in an increase of IE expression and only low levels of other viral gene products.

**HCMV-infected CD14$^+$ monocytes are targets for T cells upon VUN100bv treatment**. Since HCMV-positive donors have a high frequency of IE-specific CTLs that likely limit virus dissemination from sporadic reactivation events[5], we evaluated whether the VUN100bv-induced partial reactivation of infected CD14$^+$ monocytes would allow clearance of these cells by HCMV-specific

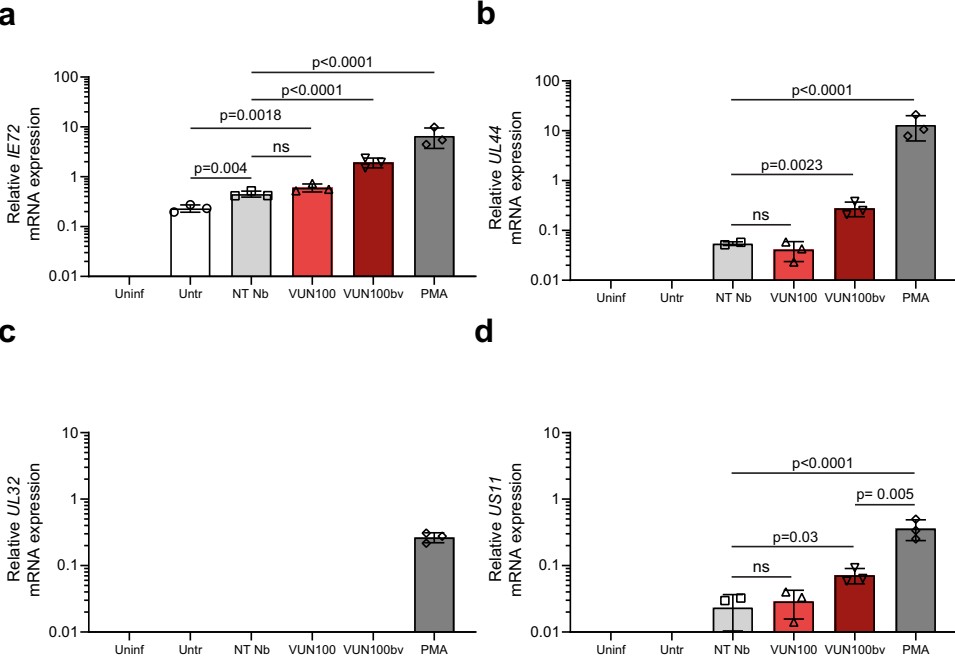

**Fig. 3 VUN100bv increases immediate-early, early and late gene but no true late gene expression.** CD14+ monocytes were left uninfected (Uninf) or infected with HCMV IE2-eYFP. Infected monocytes were left untreated (Untr), treated with a non-targeting nanobody (NT Nb), VUN100, VUN100bv, or pre-treated with 20 ng/ml PMA before infection (PMA). Six days post infection, RNA was isolated and IE72 (**a**), UL44 (**b**), UL32 (**c**), and US11 (**d**) gene expression was measured by RT-qPCR. Representative figures, showing technical replicates, from two biological replicates are shown. All data are plotted as mean ± S.D. For all figures, statistical analyses were performed using unpaired one-way ANOVA with Tukey's multiple comparison test. ns, $p > 0.05$. Source data are provided as a Source Data file.

CTLs. For this, CD14+ monocytes, CD4/CD8+ T cells, and T-cell-depleted PBMCs were isolated from HCMV-positive donors. Mindful of any potential non-specific nanobody effects, we considered a treatment of CD14+ monocytes with the non-targeting nanobody to be the most adequate control in the analysis of the results. Six days post infection and nanobody or PMA pre-treatment, IE-positive cells were counted. As seen previously, VUN100bv treatment and pre-treatment with PMA resulted in an increase of IE2-eYFP-expressing cells (Fig. 4a). After these 6 days, the infected CD14+ monocytes were further split into two groups for subsequent co-culturing with either T cells or T-cell-depleted PBMCs. After 48 h of co-culturing, T cells were removed and any remaining latently infected CD14+ monocytes were treated with GM-CSF/IL-4 and LPS, which is known to differentiate monocytes into functionally mDCs, a biologically relevant cell type that is known to reactivate HCMV from natural latency[21] (Fig. 4b). Interestingly, almost no IE-positive mDCs were observed upon treatment with either VUN100bv or PMA in combination with T-cell co-culturing. This was not seen after co-culturing with T-cell-depleted PBMCs, indicating a pivotal role for T cells for the removal of reactivated cells. In contrast, treatment with the non-targeting nanobody did not result in a significant T-cell-mediated decrease in IE-positive mDCs. In addition, we also assessed whether VUN100bv treatment of HCMV-infected cells also resulted in T-cell-mediated clearance after the establishment of latency. Six days post infection (and before nanobody treatment), cells were divided into two groups for nanobody treatment. Before the addition of nanobodies, no difference in IE2-eYFP-expressing cells between the different groups was observed (Supplementary Fig. 8). Next, cells were treated with non-targeting nanobody, VUN100bv, or PMA. One day post treatment, a significant increase of IE2-eYFP-expressing cells was seen upon VUN100bv and PMA treatment (Fig. 4c). The infected

CD14+ monocytes were again co-cultured with either T cells or T-cell-depleted PBMCs for 48 h after which T cells were removed and CD14+ monocytes were differentiated into mature dendritic cells to give cellular conditions conducive to viral reactivation (Fig. 4d). Two days post co-culturing of T cells with PMA-treated CD14+ monocytes resulted in a significant decrease of IE-positive CD14+ monocytes compared with co-culturing with T-cell-depleted PBMCs. Again, the non-targeting nanobody did not affect the number of IE-expressing cells significantly when co-cultured with either T cells or the T-cell-depleted PBMCs.

Although these data nicely illustrate the potential of the VUN100bv nanobody, one drawback of the HCMV IE2-eYFP virus used in the analysis is that it contains a deletion of the virus US2-US6 region. This region encodes several proteins that interfere with antigen presentation by, for example, down-regulating MHC Class I and II molecules[22–24]. Though it should be pointed out, that this HCMV IE2-eYFP virus does encode US11 (as shown by RT-qPCR, Fig. 3d), which can downregulate some MHC Class I molecules[25,26]. However, to ensure that our observations would also be recapitulated with a virus with a full complement of immune evasins, we repeated the co-culture experiments described above with the HCMV strain containing an intact US2-6 region[27], here termed HCMV-US2-6. CD14+ monocytes from HCMV-positive donors were infected with HCMV-US2-6 and treated with nanobody or PMA, or left untreated. After 6 days of treatment, we determined HCMV genome (g)DNA copy numbers to assess the effect of treatments on viral reactivation (Fig. 5a). Consistent with both our co-culturing experiments of HCMV IE2-eYFP infected CD14+ monocytes with fibroblasts (Figs. 2c, e) and transcript data (Fig. 3), we observed no increase in HCMV gDNA copy numbers upon VUN100bv treatment compared to the untreated or non-targeting nanobody-treated HCMV-infected CD14+ monocytes

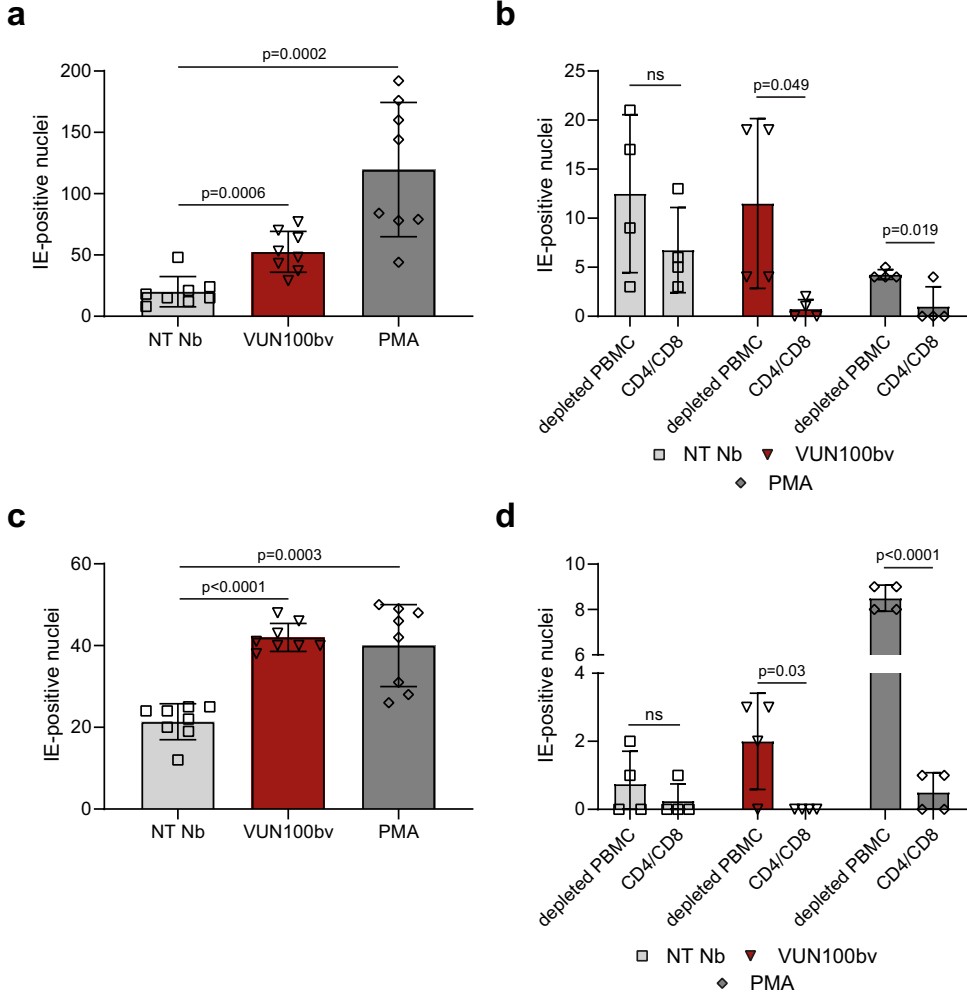

**Fig. 4 HCMV-infected CD14$^+$ monocytes are targets for HCMV-specific T cells upon VUN100bv treatment. a** Counting of IE-positive CD14$^+$ monocytes before co-culture with CD4/CD8$^+$ T cells or T-cell-depleted PBMCs. CD14$^+$ monocytes were treated with a non-targeting nanobody (NT Nb), VUN100bv for 6 days post infection or pre-treated with 20 ng/ml PMA before infection. **b** Counting of IE-positive CD14$^+$ monocytes after co-culture of T-cell-depleted PBMCs (depleted PBMC) or T cells (CD4/CD8) and differentiation of CD14$^+$ monocytes to mature dendritic cells. **c** Counting of IE-positive CD14$^+$ monocytes before co-culture with CD4/CD8$^+$ T cells or T-cell-depleted PBMCs. Six days post infection, CD14$^+$ monocytes were treated with NT Nb, VUN100bv, or 20 ng/ml PMA for 1 day. **d** Counting of IE-positive CD14$^+$ monocytes after co-culture of T-cell-depleted PBMCs (depleted PBMC) or T cells (CD4/CD8) and differentiation of CD14$^+$ monocytes to mature dendritic cells. Representative figures, showing technical replicates, from two biological replicates are shown. All data are plotted as mean ± S.D. For all figures, statistical analyses were performed using unpaired two-tailed $t$ test. ns, $p > 0.05$. Source data are provided as a Source Data file.

showing that VUN100bv treatment indeed did not result in the viral DNA replication. In contrast, PMA treatment resulted in a significant increase of HCMV gDNA copy numbers (indicating extensive viral replication). Next, untreated or nanobody-treated HCMV-US2-6 infected CD14$^+$ monocytes were co-cultured with autologous T cells for 2 days. After the removal of the T cells, HCMV gDNA copy numbers were determined prior to and after differentiation to mDCs and co-culturing with fibroblasts for 8 days to assess the reactivation potential of remaining latently infected CD14$^+$ monocytes (Fig. 5b). We saw a clear increase in HCMV gDNA copy numbers for both the untreated and non-targeting nanobody-treated infected CD14$^+$ monocytes, indicating reactivation of HCMV. In contrast, no increase of HCMV gDNA copy numbers was observed after VUN100bv treatment, consistent with loss of the latent cell pool. Overall, these data indicate that partial reactivation of latently infected cells by VUN100bv is sufficient to induce IE expression to levels that allow HCMV-specific T-cell-mediated clearing of latently infected monocytes.

## Discussion

HCMV establishes a latent infection in CD34$^+$ progenitor cells and CD14$^+$ monocytes[28,29]. Although only present in a small percentage of these cells, reactivation of HCMV can lead to disease or mortality in immunosuppressed transplant patients and immunocompromised individuals[30,31]. Importantly, no current antiviral agents target this latent reservoir, leaving an unmet need to target these latently infected cells. In this study, we set out to target the HCMV-encoded chemokine receptor US28 using a newly developed bivalent US28 nanobody to partially reactivate HCMV latently infected cells and drive their T-cell recognition and killing.

We used the previously described US28-targeting nanobody VUN100 to develop a new bivalent format VUN100bv[14]. Interestingly, the coupling of two antagonistic monovalent VUN100 nanobodies resulted in a bivalent format with partial inverse agonistic properties. Although the mechanism behind this still has yet to be explained, similar observations were made with other nanobodies targeting chemokine receptors[32–34]. VUN100bv

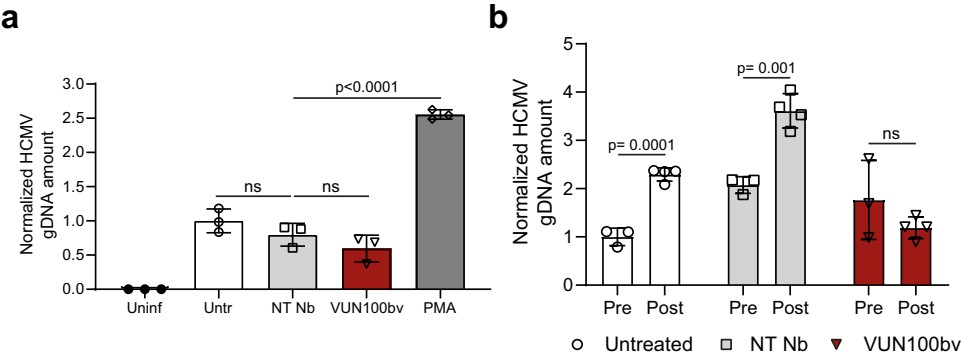

**Fig. 5 VUN100bv treatment drives a T-cell-mediated reduction of HCMV reactivation without driving viral DNA replication in latently infected cells. a** Quantification of HCMV gDNA copy numbers of uninfected (Uninf) or HCMV-US2-6 infected CD14+ monocytes before co-culture with CD4/CD8+ T cells. Infected monocytes were untreated (Untr), treated with a non-targeting nanobody (NT Nb), VUN100bv, or 20 ng/ml PMA (PMA) for 6 days. Total genomic DNA was harvested and HCMV gDNA copy numbers were determined by quantifying *GAPDH* and *UL44* gene numbers via qPCR. HCMV gDNA copy numbers were normalized to the untreated infected wells. **b** Quantification of HCMV gDNA copy numbers of untreated NT Nb or VUN100bv treated HCMV-US2-6 infected CD14+ monocytes after co-culturing with CD4/CD8+ T cells. HCMV gDNA copy numbers were determined prior to (Pre) and after (Post) differentiation of CD14+ monocytes to mature dendritic cells and co-culturing with fibroblasts for 8 days. Total genomic DNA was harvested and HCMV gDNA copy numbers were determined by quantifying *GAPDH* and *UL44* gene numbers via qPCR. HCMV gDNA copy numbers were normalized to the untreated wells prior to differentiation to mature dendritic cells. Representative figures, showing technical replicates, from three biological replicates are shown. All data are plotted as mean ± S.D. For all figures, statistical analyses were performed using unpaired two-tailed *t* test. ns, p > 0.05. Source data are provided as a Source Data file.

could bind and partially inhibit US28 signaling in cultured cell lines, and we recapitulated our observations in experimental latency settings in primary CD14+ monocytes. Here, we saw upregulation of *IE* gene expression without substantial immune evasin gene expression, late gene expression, viral DNA replication, or full virus reactivation. This is a major advantage for a shock-and-kill strategy, which requires detection by host immunity[6,35]. On a molecular level, our findings suggest that there is a threshold of inhibition of US28 signaling required for full viral reactivation.

Interestingly, the monovalent nanobody VUN100 was able to partially restore IFI16 levels in THP-1 cells and also induced some IE gene expression in CD14+ monocytes of most donors despite not inhibiting US28 constitutive signaling. VUN100 blocks ligand binding, and the ligand-binding activity of US28 has been shown to be required for the establishment in some, but not all, experimental latency systems[11,12,14,36]. Furthermore, the donor-to-donor variability we observed in the VUN100 effect raises questions about the role of US28 ligand binding during HCMV latency in patients. In contrast, VUN100bv consistently and robustly induced IE expression in all donors tested and led to recognition and killing by CTLs from seropositive individuals. Moreover, we were able to validate these results upon establishment of latency ensuring that VUN100bv not only hampers latency establishment but also induces partial reactivation of latently infected cells. Because our results were obtained using experimentally latently infected cells, follow-up experiments using ex vivo PBMCs from naturally infected donors would provide valuable confirmation of the current results. Nevertheless, our current studies are consistent with previous reports of shock-and-kill strategies using epigenetic modifiers, such as HDAC inhibitors, to induce transient viral gene expression[7,37–39]. However, treatment with HDAC inhibitors is associated with substantial off-target effects due to other and more physiological functions of these enzymes[40]. In contrast to such inhibitors, we have, here, developed a molecule specific to HCMV-infected cells and, therefore, it should display limited off-target effects. Recently, the first FDA-approved nanobody has entered the clinic for treatment of acquired Thrombotic Thrombocytopenic Purpura[41,42]. This paves the way for the potential therapeutic use

of HCMV-specific nanobodies. In addition to their intrinsic activity, the efficacy of such nanobodies in experimental and clinical settings could be further enhanced through coupling to effector molecules[14,43–47].

Overall, our study provides a strong basis for using inverse agonistic/inhibitory anti-US28 nanobodies to reduce latent viral loads in transplant donors and/or recipients prior to surgery and immunosuppression. This could lead to a lower incidence of CMV-associated disease and mortality during life-saving solid organ and stem cell transplantation.

## Methods

**Cell culture and virus infection.** Primary CD14+ monocytes were isolated from apheresis cones (NHS Blood and Transfusion Service, United Kingdom), using Lymphoprep (STEMCELL Technologies, Vancouver, Canada) density gradient centrifugation followed by magnetic-activated cell sorting (MACS) separation using CD14 microbeads (Miltenyi Biotec, Bergisch Gladbach, Germany). The monocytes were adhered to tissue culture dishes (Corning, Tewksbury, MA, USA) and were cultured in X-vivo 15 (Lonza, Walkersville, MD, USA) supplemented with 2 mM L-glutamine (Gibco, ThermoFisher Scientific, Waltham, MA, USA) at 37 °C in 5% CO₂. PMA (Sigma-Aldrich, Saint-Louis, MO, USA) was used as described in figure legends at 20 ng/mL to induce differentiation of monocytes to a macrophage-like phenotype and is known to result in reactivation of HCMV lytic infection within 24 h of treatment[13]. Primary CD14+ monocytes for use in T-cell co-culturing experiments were isolated from peripheral blood of HCMV-positive donors as for apheresis cones. CD4+ and CD8+ T cells from these HCMV-seropositive donors were isolated by MACS from monocyte-depleted PBMC using CD4 and CD8 microbeads (Miltenyi Biotec).

THP-1 cells (ATCC TIB-202), lentivirally transduced with different US28 constructs, have been described previously[12] and were cultured according to ATCC standards (RPMI-1640 media (Sigma-Aldrich)) supplemented with 10% heat-inactivated fetal bovine serum (FBS; PAN Biotech, Aidenbach, Germany), 100 U/mL penicillin and 100 µg/mL streptomycin (Sigma-Aldrich), and 0.05 mM 2-mercaptoethanol (Gibco) maintained at 37 °C in 5% CO₂.

HEK293T cells (ATCC CRL-11268) were grown at 5% CO₂ and 37 °C in Dulbecco's Modified Eagle's Medium (DMEM, ThermoFisher Scientific) supplemented with 1% Penicillin/Streptomycin (ThermoFisher Scientific) and 10% FBS (ThermoFisher Scientific). These cells were used to generate HEK293T cells overexpressing US28 and have been described previously[48].

Human foreskin fibroblasts (Hff1; ATCC SCRC-1041) were maintained in DMEM (Sigma-Aldrich) supplemented with 10% heat-inactivated FBS and 100 U/mL penicillin and 100 µg/mL streptomycin.

Viral isolate RV1164 (HCMV TB40/E strain with an IE2-eYFP tag) has been described previously[49]. Titan WT and ΔUS28 strains have been described previously[50]. For experiments using HCMV containing an intact US2-6 region, we used isolate TB40-UL32-GFP-HCMV, which has been described earlier[27]. CD14+

monocytes were infected at a multiplicity of infection of 3 for 2 h, were washed twice with phosphate-buffered saline (PBS), before replacing with fresh X-vivo 15 + L-glutamine.

**Nanobody production.** Nanobody gene fragments were recloned in a frame with a myc-His6 tag in the pET28a production vector. Bivalent formats of VUN100 were constructed by the addition of a 30GS linker in frame with the nanobody fragments. Transformed BL21 + E. coli were grown in an orbital shaker at 37 °C in the Terrific Broth containing 50 µg/mL kanamycin. When the culture reached an OD600 of 0.5, nanobody production was induced by the addition of isopropyl-$\beta$-D-thiogalactopyranoside (Sigma-Aldrich) to a final concentration of 1 mM. Incubation then continued at 37 °C for 3–4 h. Cultures were spun down for 30 min at 4000 RPM and the pellets were frozen overnight at −20 °C. The next day, pellets were thawed and resuspended in PBS. The resuspended pellet was incubated at 4 °C head-over-head at 20 RPM for 2 h. Cultures were spun down for 20 min at 4000 RPM at 4 °C and the nanobodies were purified from the supernatant using a 1 mL HisTrap HP column (GE Healthcare, Chicago, Illinois, USA). The purity of the nanobodies was verified by sodium dodecyl sulfate-polyacrylamide gel electrophoresis (SDS-PAGE) (Bio-Rad, Hercules, CA, USA).

**Nanobody binding enzyme-linked immunosorbent assay (ELISA).** Nanobody binding was performed as described previously[14]. In brief, US28-expressing membrane extracts were coated in a 96-well MicroWell MaxiSorp flat bottom plate (Sigma-Aldrich) overnight at 4 °C. The next day, wells were washed and blocked with 2% (w/v) skimmed milk (Sigma-Aldrich) in PBS. Different concentrations of nanobodies were incubated. Nanobodies were detected with mouse-anti-Myc antibody (1:1000, clone 9B11, Cell Signaling Technology, Leiden, The Netherlands) and Goat anti-Mouse IgG-HRP conjugate (1:1000, #1706516, Bio-Rad). Optical density was measured at 490 nm with a PowerWave plate reader (BioTek, Winooski, VT, USA). Data were analyzed using GraphPad Prism version 8.0 (GraphPad Software, Inc., La Jolla, CA, USA).

**Competition binding.** Membrane extracts of HEK293T and HEK293T over-expressing US28 were used during competition binding studies. Cells were washed with cold PBS and resuspended afterward in cold PBS. The cell solution was pelleted via centrifugation at $1500 \times g$ at 4 °C. The cell pellet was washed with cold PBS and spun down again. The pellet was resuspended in membrane buffer (15 mM Tris-Cl, 0.3 mM EDTA, 2 mM MgCl$_2$, pH 7.5) and disrupted by the homogenizer Potter-Elvehjem at 1200 rpm. Protein concentrations were determined using Pierce BCA protein assay kit (ThermoFisher Scientific). Increasing amounts of nanobodies or unlabeled CX3CL1 were added 100 pM $^{125}$I-CX3CL1 in HEPES binding buffer (50 mM HEPES-HCl, pH 7.4; 1 mM CaCl$_2$; 5 mM MgCl$_2$; 0.1 M NaCl; 0.5% (w/v) bovine serum albumin). Radioligand alone and radioligand + 100 nM of unlabeled CX3CL1 were used as controls for total binding and non-specific binding. A total of 3 µg of HEK293T or HEK293T overexpressing US28 were added to the ligands and incubated for 2 h at room temperature. Membranes were harvested on 0.5%(w/v) polyethylenimine (PEI)-soaked GF/C filter plates (Perkin-Elmer, Waltham, MA, USA) and dried for 30 min at 60 °C. In addition, 100 pM of $^{125}$I-CX3CL1 was spotted on the GF/C filter plate to determine radioligand concentration. Scintillation fluid MicroScint-O (Perkin-Elmer) was added to the GF/C filter plate and radio-active decay was measured using a Microbeta liquid scintillation counter (Perkin-Elmer). Data were analyzed using GraphPad Prism version 8.0.

**NFAT reporter gene assay.** HEK293T cells were detached using Trypsin-EDTA 0.05% (Gibco). In all, $1 \times 10^6$ cells were transfected with a total of 2 µg DNA and 12 µg 25 kDa linear PEI (Sigma-Aldrich) in 150 mM NaCl solution. For the transfection, 20 ng of pcDEF3-HA-US28 VHL/E WT/Y16F/R129A or 200 ng HA-US28 pcDEF3-HA-US28 ΔN(2–22) and 1 µg of NFAT-luciferase reporter gene (Stratagene, La Jolla, CA, USA) was used, which was supplemented with pcDEF3 empty DNA to a total of 2 µg. The DNA-PEI mixture was vortexed for 3 s and incubated for 15 mins at room temperature and resuspended in DMEM. The HEK293T cell suspension was added to DNA-PEI mixture and 30,000 cells per well were seeded in a white poly-L-lysine coated 96-wells plate. Six hours post-transfection, nanobodies were added with a final concentration of 100 nM or 1 µM, and cells were incubated at 37 °C and 5% CO$_2$. After 24 h, supernatant was removed and 25 µL LAR (0.83 mM D-luciferine, 0.83 mM ATP, 0.78 µM Na$_2$HPO$_4$, 18.7 mM MgCl$_2$, 38.9 mM Tris-HCl (pH 7.8), 2.6 µM DTT, 0.03% Triton X-100 and 0.39% Glycerol) was added. Luminescence (3 s per well) was measured using a VICTOR$_3$ multilabel plate reader (Perkin-Elmer). Raw data were normalized to the average signal of the wells containing untreated US28 WT cells. Data were analyzed using GraphPad Prism version 8.0.

**US28 receptor expression ELISA.** Transiently transfected cells, used for NFAT reporter assay, were seeded at 50,000 cells per well in poly-L-lysine coated 96-well plates and were grown at 37 °C and 5% CO$_2$. The next day, cells were fixed with 4% paraformaldehyde (Sigma-Aldrich) for 10 min at room temperature. Cells were permeabilized with 0.5% NP-40 (Sigma-Aldrich) for 30 min at room temperature and subsequently blocked for 30 min at room temperature in 1% (v/v) FBS/PBS. Cells were incubated with the rat-anti-HA antibody (1:1000, Clone 3F10, Roche,

Basel, Switzerland) for 1 h at room temperature. Subsequently, cells were incubated with Goat anti-Rat IgG-HRP conjugate (1:1000, Pierce, Thermo Scientific) for 1 h at room temperature. Between all incubation steps, cells were washed three times with PBS. 1-Step Turbo TMB-ELISA substrate (Thermo Scientific) was added to the wells and the reaction was stopped with 1 M H$_2$SO$_4$. Optical density was measured at 450 nm with a PowerWave plate reader. Data were analyzed using GraphPad Prism version 8.0.

**Immunofluorescence microscopy.** THP-1 cells were spun down at $500 \times g$ for 5 min, resuspended in 4% paraformaldehyde (Sigma-Aldrich), and seeded in a 96-well U-bottom plate. Cells were fixed for 10 mins at room temperature. After fixation, cells were permeabilized with 0.5% NP-40 (Sigma-Aldrich) for 30 min at room temperature. Nanobodies were incubated for 1 h at RT and detected using Mouse-anti-Myc antibody (1:1000, 9B11 clone, Cell Signaling). US28 was visualized with the rabbit-anti-US28 antibody (1:1000, Covance, Denver, PA, USA[51]) for 1 h at room temperature. Subsequently, cells were washed and incubated with Goat anti-Rabbit Alexa Fluor 546 (1:1000 in 1% (v/v) FBS /PBS, ThermoFisher Scientific) and Goat anti-Mouse Alexa Fluor 488 (1:1000 in 1% (v/v) FBS/PBS, ThermoFisher Scientific).

IE antigen was detected in monocytes and fibroblasts by fixation and permeabilization in 70% ethanol at −20 °C for 30 mins and blocking using PBS with 1 % bovine serum albumin and 5% goat serum, and then incubating with mouse-anti-IE antibody (1:1000, #11-003, Argene, bioMérieux, Marcy-l'étoile, France) followed by secondary antibodies as described above.

**Western blot.** Mock transduced or US28-expressing THP-1 cells were seeded in a six wells plate and incubated with 100 nM nanobodies. After 48 h, cells were lysed in native lysis buffer (25 mM Tris-HCL pH 7.4, 150 mM NaCl, 1 mM EDTA, 1% NP-40, 5% Glycerol, 1 mM NaF, 1 mM NaVO$_3$, cOmplete protease inhibitor cocktail) for 10 min on ice. Cell debris was removed by centrifugation at $13,000 \times g$. Protein concentration of lysates was determined by Pierce BCA protein assay kit (ThermoFisher Scientific) and the same protein quantities were separated on a 10% SDS-PAGE gel under reducing conditions and transferred to 0.45 µm poly-vinylidene fluoride blotting membrane (GE healthcare, Chicago, IL, USA). Total ERK1/2 and phospho-ERK1/2 were detected using p44/42 MAPK antibody (1:1000 in 5% BSA/TBS-T, #9102, Cell Signaling) and phospho-p44/42 MAPK (Thr202/Tyr204) (1:1000 in 5% BSA/TBS-T, #9106, Cell Signaling). Total IFI16 was detected using anti-IFI16 antibody (1:500 in 5% BSA/TBS-T, sc-8023, Santa Cruz). Actin was detected using anti-actin antibody (1:2000 in 5% BSA/TBS-T, Clone AC-74, Sigma-Aldrich). Antibodies were detected using Goat anti-Rabbit IgG-HRP conjugate (1:10000, #1706515, Bio-Rad) or Goat anti-Mouse IgG-HRP conjugate (1:10000, #1706516, Bio-Rad). Blots were developed using Western Lightning Plus-ECL (Perkin-Elmer) and visualized with Chemidoc (Bio-Rad).

**Detection of IE expression and IE focus formation.** CD14$^+$ monocytes were isolated and seeded in a 96-well plate. In some experiments, as a positive control, CD14$^+$ monocytes were pre-treated with 20 ng/ml PMA 1 day after seeding. The next day, the medium was removed and cells were infected with RV1164 viral isolate. Two hours post infection, the medium was aspirated and replaced with medium containing nanobodies at a final concentration of 100 nM. Three days post infection, nanobody-containing medium was refreshed. Six days post infection, IE expression was detected by means of IE2-eYFP tag or staining of IE as described above. These were either counted manually or using the Target Activation experimental tool of the ArrayScan XTI instrument (ThermoFisher), using Hoechst stained nuclei for object identification.

During the prolonged protocol, to allow the establishment of latency, CD14$^+$ monocytes were isolated and seeded in a 96-wells plate. The next day, the medium was removed and cells were infected with RV1164, Titan WT, or Titan ΔUS28 viral isolate. Two hours post infection, the medium was aspirated and media was replaced with fresh X-vivo 15 + L-glutamine. Six days post infection, IE-expressing cells were counted. Next, the medium was aspirated and replaced with medium containing nanobodies at a final concentration of 100 nM of PMA at a final concentration of 20 ng/ml. IE expression was determined two days after the addition of nanobodies or PMA. Next, Hff1 cells were detached using Trypsin-EDTA 0.05% (Gibco). The medium of the wells containing CD14$^+$ monocytes was removed and 10,000 Hff1 cells were co-cultured with the CD14$^+$ monocytes to determine the production of infectious viral particles. IE focus formation was monitored for up to 14 days.

**RNA extraction and analysis.** CD14$^+$ monocytes were isolated and infected as described above. Six days post infection, cells were washed once with 1× PBS, and RNA was harvested by adding Trizol reagent (Zymo Research, Irvine, CA, USA). RNA was isolated using Direct-Zol RNA MiniPrep kit (Zymo Research) according to the manufacturer's instructions. cDNA was produced by Quantitect Reverse Transcription kit (Qiagen, Hilden, Germany) according to the manufacturer's instructions.

qPCR was performed using LUNA SYBR green qPCR reagents (New England Biolabs, Ipswich, MA, USA) using primers presented in Supplementary Table 2. Viral transcript levels were normalized to GAPDH and are presented as $2^{\Delta Ct}$.

**PBMC and T-cell co-culture and virus reactivation**. For experimental latency experiments, following PBMC isolation from an HCMV-positive healthy donor peripheral blood, CD14+ monocytes were isolated, plated on 96-well plates, and treated as described above and in the figure legends, and the remaining PBMC were frozen in liquid nitrogen until one day prior to co-culture. At this time, the PBMC were thawed and rested overnight. CD4+ and CD8+ T-cell fractions were isolated as above and pooled, and these, or the remaining PBMC were added to the monocyte cultures at an effector:target cell ratio of 5:1. After 2 days, the T cells/depleted PBMC were washed away using PBS + 2 mM EDTA, and the medium on the monocytes was replenished with X-vivo 15 + L-glutamine containing interleukin-4 (Miltenyi Biotec) and granulocyte-macrophage colony-stimulating factor (Miltenyi Biotec) at 1,000 U/ml in order to stimulate differentiation to immature dendritic cells, along with 10 μg/mL anti-HLA-A, B, C (Biolegend) and 10 μg/mL anti-HLA-DR, DP, DQ (BD Bioscience) to block any further T-cell killing. After 5 days, this medium was aspirated and replaced with X-vivo 15 + L-glutamine supplemented with 50 ng/mL LPS (Invivogen, San Diego, CA, USA) for 2 days to induce maturation of dendritic cells, a biologically relevant cell type which is known to reactivate HCMV from natural latency[21]. This media was aspirated and $1 \times 10^4$ Hff1 cells were added to each well in DMEM-10 medium. IE focus formation was determined during a period of 10 days of co-culturing.

**Detection of HCMV genomic DNA in latent and reactivated CD14+ monocytes**. CD14+ monocytes were isolated from an HCMV-seropositive individual and seeded in a 48-well plate. The next day, the medium was removed and cells were infected with the HCMV-US2-6 (TB40-UL32-GFP) viral isolate. Two hours post infection, the medium was aspirated and replaced with medium containing nanobodies at a final concentration of 100 nM, or PMA at 20 ng/mL. Three days post infection, nanobody-containing medium was refreshed. Six days post infection, ~1/3 of the wells were harvested for DNA extraction (described below). The remaining wells were treated with CD4/CD8+ T cells in the presence of nanobodies for two days before removal as described above. Approximately 1/3 of the wells were harvested for DNA extraction and the remaining wells were treated with X-vivo 15 + L-glutamine medium containing interleukin-4 (Miltenyi Biotec) and granulocyte-macrophage colony-stimulating factor (Miltenyi Biotec) at 1,000 U/ml in order to stimulate differentiation to immature dendritic cells, along with 10 μg/mL anti-HLA-A, B, C (Biolegend) and 10 μg/mL anti-HLA-DR, DP, DQ (BD Bioscience) to block any further T-cell killing. After 5 days, this medium was aspirated and replaced with X-vivo 15 + L-glutamine supplemented with 50 ng/mL LPS (Invivogen, San Diego, CA, USA) for 2 days to induce maturation of dendritic cells. This media was aspirated and $3 \times 10^4$ Hff1 cells were added to each well in DMEM-10 medium. After 7–8 days, these wells were harvested for DNA extraction as follows.

For analysis of viral genomes, cells were washed with PBS, then solution A (100 mM KCl, 10 mM Tris-HCl pH8.3 2.5 mM MgCl) was added followed by an equal volume of solution B (10 mM Tris-HCl pH8.3, 2.5 mM MgCl 1% Tween20, 1% NP-40, 0.4 mg/ml proteinase K). Cells were scraped and placed in microtubes before heating at 60 °C for 1 h then 95 °C for 10 mins. Two μl of each solution was used in qPCR analysis of the UL44 non-transcribed promoter region, using the GAPDH non-transcribed promoter region to correct for total DNA levels. Primer sequences are provided in the table.

**Ethical approval for the use of human samples**. Human samples were obtained under ethical approval from Cambridgeshire 2 Research Ethics Committee (REC reference 97/092) conducted in accordance with the Declaration of Helsinki. All volunteers have given informed written consent before providing blood samples.

**Statistical analysis**. For the validation of the nanobody, each experiment is consisting of two to four technical replicates of three or four biological replicates. For the virus experiments, three to six technical replicates of one to four different donors have been used. No outliers were removed from the experiments. Data were analyzed using GraphPad Prism version 8.0 software. All figures are representative figures and all data are plotted as mean ± S.D. Statistical analyses were performed using unpaired two-tailed $t$ test, one-way ANOVA with Tukey's multiple comparison tests, two-way ANOVA with Tukey's multiple comparison test, or the Holm–Sidak method with alpha = 0.05 and are mentioned in the figure legends. $P$ value < 0.05 was considered statistically significant.

**Biological material availability**. US28-targeting nanobodies, described in this paper, can be obtained through an MTA. The HCMV strains RV1164 (TB40-IE2-eYFP-HCMV), Titan WT, Titan ΔUS28, and TB40-UL32-GFP-HCMV are all published and not ours to make available.

**Reporting summary**. Further information on research design is available in the Nature Research Reporting Summary linked to this article.

## Data availability

All data associated with this study are present in the paper or in the Supplementary Information file. The data that support the findings of this study are available from the corresponding author upon reasonable request. Source data are provided with this paper.

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

## Acknowledgements

We thank Linda Teague, Roy Whiston, Georgina Brown, and Veronika Romashova for technical assistance. This work was supported by the Dutch Research Council (NWO: Vici grant 016.140.657), the Wellcome Trust (Grant 109075/Z/15/A), the British Medical Research Council (grant MR/K021087/1), and Cambridge NIHR BRC Cell Phenotyping Hub.

## Author contributions

Conceptualization: T.D.G., E.E., R.H., J.S., and M.S.; investigation: T.D.G, E.E., E.L., I.G., and N.B.; formal analysis: T.D.G. and E.E.; writing-original draft preparation: T.D.G. and E.E; writing-review and editing: R.H., M.W., J.S. and M.S.; visualization: T.D.G. and E.E; supervision: R.H., M.W., J.S. and M.S.; funding acquisition: J.S., M.W., and M.S.

## Competing interests

The authors declare the following competing interests: R.H. is affiliated with QVQ Holding BV, a company offering VHH services and VHH-based imaging molecules. T.D.G, E.E., R.H., J.S., and M.S. are co-inventors on a pending EU patent application EP19190047.1, which covers the use of US28-targeting nanobodies for T-cell-mediated killing of HCMV latently infected cells. The remaining authors declare no competing interests.
