## [Peer Review File · Nature Communications]

Parts of this Peer Review File have been redacted as indicated to remove third-party material where no permission to publish could be obtained.

REVIEWER COMMENTS

Reviewer #1 (Remarks to the Author):

In this manuscript by DeGroof et al, entitled "Targeting the latent human cytomegalovirus reservoir with virus specific nanobodies", the authors detail the development of a bivalent nanobody specific to the cytomegalovirus (CMV) protein US28, which could perhaps be used to eliminate latent cells from donor transplant material. This would represent a significant advancement in the transplant field, particularly for bone marrow transplants, so this work is quite timely. Overall, the authors have asked the right questions. However, certain aspects of the work must be addressed. Specific suggestions are provided.

Major

1. Figure 1. Panels A-C are done in HEK cells using US28 overexpression. In C, the authors test US28 constitutive activity by assessing NFAT. However, to demonstrate this is indeed mediated by US28 constitutive signaling, US28 mutant expression constructs should be used to demonstrate NFAT is modulated in the absence of a ligand. (It is also not clear if there is precedent that US28 constitutively regulates NFAT, as no reference is cited to support this notion). The authors conclude, "By competing for the endogenous ligands of US28...partial inverse agonist." However, they have only tested one of US28's many ligands. Further, this should be performed with signaling mutant constructs (e.g. US28-R129A, US28-deltaN, US28-Y16F), which would solidify such a conclusion. Proper controls are also needed for all panels (e.g. Panels C-D need empty vector controls to show effects are US28 specific; Panel D additionally needs untreated controls).
2. A problem with the overall interpretation of the data is that latency is established, and the VUN100b nanobody "reverses latency", but cannot fully reactivate virus. However, I do not think the authors can conclude this given their methods. The assumption here is that latency was established in the presence of the nanobody in the first place. The authors infect cells for 2h, and then add fresh medium containing nanobodies and perform assays at 2 and 6dpi. US28 is required for establishment of latency, which is arguably still occurring at 2hpi. Thus, if the argument is that these nanobodies inhibit US28 signaling, and US28 is required for latency establishment, what is to say the nanobodies are blocking the establishment of latency in the first place? There are no experiments included to show latency was indeed established to equal levels as untreated and Irr Nb latently infected cells. Nonetheless, a better approach is to allow latency to establish for 7-10days, then add the nanobodies and ask if virus can reactivate. This certainly more closely resembles what would happen in vivo, as well.
3. Figure 2: A control is needed in each panel. The authors use CD14 monocytes PRE-treated with PMA as their "control", but this is not a good control, as such cells favor lytic, not latent, infection. A suitable control is CD14 monocytes latently infected and THEN treated with PMA to induce reactivation. This goes hand-in-hand with the above comment, because this could simply be a failure to establish latency, and have nothing to do with reactivation at all. Hence, a reactivation control is necessary. It is unclear why A shows 2dpi and B shows 6dpi. All data should be shown and matched. In panel C, it is unclear how "plaque formation" is quantified? Further, these time points do not seem accurate for a latency/reactivation assay. After only 6 days (which is barely enough time to measure latency maintenance, the authors coculture the CD14 cells with Hff1 cells, and then quantify plaques after only 4 days. Most reactivation assays go for 10-14days (or longer). Are the authors really counting plaques? Or are they counting IE-YFP clusters (which are not plaques)? There are no details on these methods, and these assays do not seem to be performed correctly or with the appropriate controls to properly evaluate reactivation.
4. Figure 3: Proper controls are missing. All panels need infected/non-treated controls. Further, PMA needs to be added as controls after latency is established (not prior to infection). deltaUS28 infected cells + inhibitor should be shown as a control to demonstrate specificity for US28 in the context of infection.
5. Figure 4: Proper controls are missing. All panels need infected/non-treated controls. A positive control showing latently infected, reactivated cultures is also important. These data are highly qualitative, and are solely depended on IE protein expression. This does not measure "reactivation", or even lytic infection, without additional experimentation. IE gene/protein does not always translate to infectious virus (even in lytic infection). If the authors are going to test "full reactivation" as they indicate, then they need to perform an experiment that measures much more than IE-positive nuclei.

6. Supplemental Fig 4: It is difficult to understand how the authors now detect so much UL32 positivity, when their transcript data presented in Fig 3C in each condition shown in Fig S4 is 'n.d.' in Fig 3C. Arguably neither dataset contains the important infected/untreated control, but given the number of UL32-positive cells they detect by fluorescence (A), it is hard to imagine there are no transcripts detected (fig3c). This figure also needs latently infected, non-treated +/- PMA controls.
7. Statistical analysis: "All figures are representative figures..." If this is the case, then all figure legends have to explicitly state the number of biological replicates performed per experiment (as the authors say 3-6 in this methods section). It is unclear as to why biological replicate data was not combined.

Minor

1. Clarity to experimental design is warranted
2. Clarity to premise for experiment (is NFAT constitutively activated by US28? Where is the reference?)
3. Abstract: what do the authors mean by "VUN100b treatment partially reverses latency without fully reactivating the virus"? Specifically, the authors need to clarify what "reverses latency" means.
4. Introduction: "often with little-to-no viral gene expression" – I would encourage the authors to edit this phrase, as there are indeed viral genes expressed during latency.
5. Introduction: Is CMV really a "leading cause of transplant rejection"?
6. Introduction: "suppress IE gene expression via the repression of chromatin structure at the viral MIEP". This should be reworded. Repressive chromatin marks are involved, but stating 'repression of chromatin structure' is inaccurate. Also, multiple transcription factors are involved in this as well, which are addition to the chromatin remodeling. Finally, these alterations are, for the most part, in the MIE enhancer, not the MIEP.
7. Figure 1E: Are the stats from multiple western analyses or multiple measurements of the single blot?
8. Supplementary figure numbers are missing in the methods – I believe they are to reference Fig S3.
9. Biological material availability section: 2nd sentence – should TB40-UL32-GFP-CMV read "IE2-YFP" instead?

Reviewer #2 (Remarks to the Author):

In this manuscript authors report in vitro testing of a virus specific nanobody (VUN100b) that partially inhibits signaling of the viral receptor US28. Using an in vitro latency model of CMV, authors demonstrate that VUN100b treatment partially reverses latency without fully reactivating the virus. Furthermore, they also show that VUN100b treatment drives recognition and killing of latently infected monocytes by autologous CMV-specific T cells from HCMV-seropositive individuals. Based on these observations, authors propose that VUN100b can be potentially used as a therapy to clear the HCMV latent reservoir of transplant patients. Overall this is an interesting preliminary observation which is strongly supported by the data presented in this manuscript. However, lack of any data on the assessment of an in vivo efficacy makes it very difficult to evaluate potential use of VUN100b. There are few previously published reports on latent CMV infection in humanized murine model. I was wondering if authors have considered using one of these models to assess the therapeutic potential of VUN100b. I hope authors would agree that there are many therapeutic drugs which show high efficacy in in vitro but often fail in a pre-clinical or clinical setting.

Reviewer #3 (Remarks to the Author):

In this paper, the authors describe the effect of a bivalent nanobody (VUN100b) that acts as a inverse agonist on its US28 target, a HCMV membrane receptor on latently infected cells. The VUN100b binding to US28 also prevents or disrupts the binding of this viral receptor to host ligands such as CX3CL1. Upon antigen interaction with VUN100b, US28 reactivates some, but not

all, genes of HCMV. Interestingly no (or very low numbers) of infective HCMV particles are produced. Surprisingly, treatment of latently infected monocytes with this bivalent nanobody in presence of CTLs from HCMV seropositive persons will severely reduce the HCMV reservoir cells. Thus, this VUN100b construct could be developed as a therapeutic tool to remove HCMV latently infected cells in transplant patients.

It is an interesting and unexpected observation, the experiments are well designed and provide convincing evidence to support the conclusions.

My expertise is directed more towards antibody engineering, rather than virology. An antibody engineer will not find much details in this paper as the reader is directed to a previous paper (De Groof et al, Mol Pharm, 2019) where this nanobody has been characterised in more detail. Here this nanobody was made bivalent to obtain an increased binding due to avidity effects. Of note, it would be better to name this bivalent nanobody VUN100bv (instead of VUN100b), this way it is more clear and a difference can be made between bivalent (bv) bispecific (bs) and biparatopic (bp) as such constructs might be preferred in the future.

Also, since this bivalent construct is more effective than the monovalent nanobody, it might be a good idea to communicate the expression yield of the bivalent construct (relative to monomer). Bivalent nanobody constructs are expected to have a lower yield, while the same effect can be observed by adding slightly larger amounts of the monovalent nanobody on the cells.

It would be very interesting to see the crystal structure of the (monomeric) nanobody in complex with SU28. However, it is conceivable that the crystallisation of this challenging protein (complex) will require an enormous effort, and might not be feasible in the end. Nevertheless, such crystal structure will provide the fine details of the exact epitope, and might be the start to understand the intriguing inverse agonist behaviour of this nanobody.

The paper is organised in a logical order and English grammar and vocabulary are appropriate.

Reviewer #1 (Remarks to the Author):

In this manuscript by De Groof et al, entitled “Targeting the latent human cytomegalovirus reservoir with virus specific nanobodies”, the authors detail the development of a bivalent nanobody specific to the cytomegalovirus (CMV) protein US28, which could perhaps be used to eliminate latent cells from donor transplant material. This would represent a significant advancement in the transplant field, particularly for bone marrow transplants, so this work is quite timely. Overall, the authors have asked the right questions. However, certain aspects of the work must be addressed. Specific suggestions are provided.

Major

1. Figure 1. Panels A-C are done in HEK cells using US28 overexpression. In C, the authors test US28 constitutive activity by assessing NFAT. However, to demonstrate this is indeed mediated by US28 constitutive signaling, US28 mutant expression constructs should be used to demonstrate NFAT is modulated in the absence of a ligand. (It is also not clear if there is precedent that US28 constitutively regulates NFAT, as no reference is cited to support this notion). The authors conclude, “By competing for the endogenous ligands of US28...partial inverse agonist.” However, they have only tested one of US28’s many ligands. Further, this should be performed with signaling mutant constructs (e.g. US28-R129A, US28-deltaN, US28-Y16F), which would solidify such a conclusion. Proper controls are also needed for all panels (e.g. Panels C-D need empty vector controls to show effects are US28 specific; Panel D additionally needs untreated controls).

We thank the reviewer for the valuable feedback on our manuscript and finding our results an advancement in particular in the transplant field. We agree with reviewer 1 that the clarity of some of the figure panels could be aided by the addition of controls. We have included the literature reference in the text that describes that US28 constitutively modulates NFAT activation. Moreover, we have performed additional experiments testing signaling of 3 different US28 mutants (US28 Δ N mutant, US28 Y16F mutant and US28-R129A mutant) and the effect of the inverse agonistic VUN100(bv) nanobody on NFAT-signaling mediated by these mutants. This new data confirmed that NFAT signaling is indeed mediated by the constitutive activity of US28 (new Figure 1C). Also, we have added mock controls (only expressing the NFAT reporter) to the figure. We have added the information regarding the normalization in the figure legend and in the material and methods section.

Figure 1B indeed only shows displacement of CX3CL1 by the bivalent nanobody format. In this displacement experiment we merely focus on the enhanced potency of the bivalent format of VUN100. In a previous study, we have characterized VUN100 in more detail and have shown that VUN100 also displaces other chemokines. To clarify the further, we have added this information (“which was previously shown to displace US28 endogenous ligands”) in the text. Moreover, we have also changed our conclusion at the end of the paragraph to “by competing for binding of CX3CL1”.

In addition, the empty vector control of Figure 1D, requested by Reviewer 1, can be found as Supplemental figure 1. We have also included the untreated control for both the empty vector transduced THP-1 cells and US28-expressing THP-1 cells to both figure 1D and supplementary figure 3. Combined, these data show that VUN100, the bivalent VUN100 and the anti-US28 antibody bind to US28 in these cells.

2. A problem with the overall interpretation of the data is that latency is established, and the VUN100b nanobody “reverses latency”, but cannot fully reactivate virus. However, I do not think the authors can conclude this given their methods. The assumption here is that latency was established in the presence of the nanobody in the first place. The authors infect cells for 2h, and then add fresh medium containing nanobodies and perform assays at 2 and 6dpi. US28 is required for establishment of latency, which is arguably still occurring at 2hpi. Thus, if the argument is that these nanobodies inhibit US28 signaling, and US28 is required for latency establishment, what is to say the nanobodies are blocking the establishment of latency in the first place? There are no experiments included to show latency was indeed established to equal levels as untreated and Irr Nb latently infected cells. Nonetheless, a better approach is to allow latency to establish for 7-10days, then add the nanobodies and ask if virus can reactivate. This certainly more closely resembles what would happen in vivo, as well.

We agree with the comment of the reviewer and have performed additional experiments to show the effect of the nanobodies on cells with already established latency (Figure 5 and supplementary Figures 7-9). To this end, we have infected CD14+ monocytes with IE2-YFP virus for 6 days to establish latency (a time frame we and other have routinely used for establishment of latent infection in primary monocytes), this time without any nanobodies. Before the subsequent nanobody treatment, we have determined the IE-expression of all wells to ensure that latency is similar (Supplementary Figure 7 and 9). Six days post infection, we have treated the infected cells with the nanobodies or PMA and counted IE-expressing cells 2 days later. After these 2 days of nanobody or PMA treatment, we co-cultured these cells with fibroblasts for an additional 8 days to check the production of infectious particles.

In line with the previous experiment, this new setup also showed that inhibition of US28 activity by the bivalent VUN100 nanobody reactivated latently infected cells without the production of infectious particles. Moreover, we have repeated this experiment using another viral strains (Titan WT and Δ US28, Supplementary Figure 8), in which nanobody treatment 6 days post infection resulted in the same effects and was shown to be US28 specific. Moreover, we have repeated our T cell killing assays (Figure 5C-E) using this setup where we also show that VUN100bv treatment results in T cell-mediated clearance of partially reactivated CD14+ monocytes after VUN100bv treatment. Overall, these independent assays after the long-term establishment of latency as requested by the reviewer, showed similar results as our previous experimental set-up, showing that the potential of VUN100bv as shock-and-kill therapy to reactivate and clear latently infected cells.

3. Figure 2: A control is needed in each panel. The authors use CD14 monocytes PRE-treated with PMA as their “control”, but this is not a good control, as such cells favor lytic, not latent, infection. A suitable control is CD14 monocytes latently infected and THEN treated with PMA to induce reactivation. This goes hand-in-hand with the above comment, because this could simply be a failure to establish latency, and have nothing to do with reactivation at all. Hence, a reactivation control is necessary. It is unclear why A shows 2dpi and B shows 6dpi. All data should be shown and matched. In panel C, it is unclear how “plaque formation” is quantified? Further, these time points do not seem accurate for a latency/reactivation assay. After only 6 days (which is barely enough time to measure latency maintenance, the authors coculture the CD14 cells with Hff1 cells, and then quantify plaques after only 4 days. Most reactivation assays go for 10-14days (or longer). Are the authors really counting plaques? Or are they counting IE-YFP clusters (which are not plaques)? There are no details on these methods, and these assays do not seem to be performed correctly or with the appropriate controls to properly evaluate reactivation.

We agree with the reviewer and have addressed this accordingly in the revised manuscript. We would like to refer to our response to comment 2 regarding the different experimental set up and controls. Figure 2A shows the results of a qualitative determination of upregulation of IE-expression at 2dpi. In addition, quantitative data is provided for 6dpi. We also agree with the reviewers’ notion regarding plaque formation. We have changed this to “IE-focus formation” accordingly. On top of that, we have performed longer co-culturing experiments using the different experimental set up discussed above (Figure 5B). We have provided additional details on these experiments in the material and method section.

4. Figure 3: Proper controls are missing. All panels need infected/non-treated controls. Further, PMA needs to be added as controls after latency is established (not prior to infection). Δ US28 infected cells + inhibitor should be shown as a control to demonstrate specificity for US28 in the context of infection.

We have added the untreated infected controls as requested by the reviewer. As indicated in our response to question 2, we did not observe any differences in outcome between experiments in which monocytes were treated with PMA prior to infection and then treated 2h post-infection with nanobody or treated with nanobody after long-term latency had been established for 6 days. Therefore, we believe in this case, the results obtained, using pre-treated PMA samples, are still valid.

Regarding the specificity of the nanobodies in a viral setting, we have performed additional experiments. We have infected CD14+ monocytes for 6 days with Titan WT and Titan- Δ US28 virus (Supplementary Figure 8). Six days post infection, cells were treated with the nanobodies or PMA for 2-3 days and IE-expression was determined. We did not observe any effect of the US28 nanobodies on IE-expression in monocytes infected with Titan- Δ US28 virus. This has now been added to the revised version of the manuscript.

5. Figure 4: Proper controls are missing. All panels need infected/non-treated controls. A positive control showing latently infected, reactivated cultures is also important. These data are highly qualitative, and are solely depended on IE protein expression. This does not measure “reactivation”, or even lytic infection, without additional experimentation. IE gene/protein does not always translate to infectious virus (even in lytic infection). If the authors are going to test “full reactivation” as they indicate, then they need to perform an experiment that measures much more than IE-positive nuclei.

We agree with the reviewer that IE expression does not translate into lytic infection or the production of infectious particles. However, we feel that the reviewer would agree that reactivation does not occur without IE expression. We would also like to stress that our experimental setup described in Figure 4 is actually based on determining both IE expression and the effect of (IE-specific) T cells. Instead of testing full viral reactivation, our particular aim was to detect IE-expressing, reactivated cells that are remaining after T-cell co-culturing. We have validated this finding using a different virus strain (UL32-GFP), in which after T cell co-culturing, we co-cultured the latently infected monocytes with fibroblasts to look at total CMV genome levels.

However, we have addressed the question of Reviewer 1 repeating our T cell killing assays (Figure 5C-E) using reactivated latently infected cells showing that VUN100bv treatment results in T cell-mediated clearance of partially reactivated CD14+ monocytes after VUN100bv treatment.

Regarding the non-treated control samples, we feel that we also show in all the previous experiments that the irrelevant nanobody can be regarded as a proper negative control. It should be noted that we sometimes do see a small effect of the irrelevant nanobody, which could be a consequence of the manipulation of the cells, addition of proteins in general or perhaps residual contaminants from the nanobody production and purification procedure. For that reason, we believe that comparing the results between the bivalent VUN100 and the irrelevant nanobody is more correct to since this will take non-specific effects into account.

6. Supplemental Fig 4: It is difficult to understand how the authors now detect so much UL32 positivity, when their transcript data presented in Fig 3C in each condition shown in Fig S4 is 'n.d.' in Fig 3C. Arguably neither dataset contains the important infected/untreated control, but given the number of UL32-positive cells they detect by fluorescence (A), it is hard to imagine there are no transcripts detected (fig3c). This figure also needs latently infected, non-treated +/- PMA controls.

In response to the comment of the reviewer, we have included the untreated infected controls to Figure 3. With respect to the differences in UL32 positivity, this difference is indeed striking. Nevertheless, we would like to stress that it is impossible to compare the results between both experiments, since these concern 2 different virus strains and different donors. Despite differences between different viral strains, we would like to note that for both TB40-IE2YFP and Titan WT virus strains, clear induction of IE-expression but no IE-focus formation was observed upon treatment of the monocytes with bivalent VUN100. This strongly suggests that the partial reactivation mediated by the nanobody seems to be consistent with different viral strains and can therefore be considered is a general mode of action in latently infected cells.

7. Statistical analysis: "All figures are representative figures...." If this is the case, then all figure legends have to explicitly state the number of biological replicates performed per experiment (as the authors say 3-6 in this methods section). It is unclear as to why biological replicate data was not combined.

We have added the number of biological replicates to the figure legends. We did not combine the biological replicates because there are different baselines of infected and reactivated cells between the different donors. This is nicely shown in supplementary figure 4. The variability between donors makes it difficult to combine biological repeats.

Minor

1. Clarity to experimental design is warranted

We have added more details regarding the determination of IE-expression and IE-focus formation in the material and method section. We've also updated other material and method paragraphs with information about the new experiments that were performed for the revised version of the manuscript.

2. Clarity to premise for experiment (is NFAT constitutively activated by US28? Where is the reference?)

The reference has been added and we have expanded the experiment using different US28 mutants to show that NFAT is constitutively activated by US28.

3. Abstract: what do the authors mean by "VUN100b treatment partially reverses latency without fully reactivating the virus"? Specifically, the authors need to clarify what "reverses latency" means.

We have changed this to "VUN100bv treatment partially reactivated latently infected cells, by inducing IE-expression, without establishing virus production."

4. Introduction: "often with little-to-no viral gene expression" – I would encourage the authors to edit this phrase, as there are indeed viral genes expressed during latency.

We agree with the reviewer and we have changed to "limited viral gene expression".

5. Introduction: Is CMV really a “leading cause of transplant rejection”?

We have changed “leading cause” to “major cause”.

6. Introduction: “suppress IE gene expression via the repression of chromatin structure at the viral MIEP”. This should be reworded. Repressive chromatin marks are involved, but stating ‘repression of chromatin structure’ is inaccurate. Also, multiple transcription factors are involved in this as well, which are addition to the chromatin remodeling. Finally, these alterations are, for the most part, in the MIE enhancer, not the MIEP. Rephrase

We have changed this to “Latently infected CD34+ hematopoietic progenitor cells and their derived CD14+ monocytes suppress IE gene expression via the remodeling of chromatin structure and multiple repressive transcription factors at the viral major immediate early promotor/enhancer region”

7. Figure 1E: Are the stats from multiple western analyses or multiple measurements of the single blot?

The stats are from biological replicates from multiple western blots. We have clarified this in the in the figure legends.

8. Supplementary figure numbers are missing in the methods – I believe they are to reference Fig S3.

The reference to supplementary figure 3 was missing. We have added this in the manuscript.

9. Biological material availability section: 2nd sentence – should TB40-UL32-GFP-CMV read “IE2-YFP” instead?

Indeed, we have changed this accordingly.

Reviewer #2 (Remarks to the Author):

In this manuscript authors report in vitro testing of a virus specific nanobody (VUN100b) that partially inhibits signaling of the viral receptor US28. Using an in vitro latency model of CMV, authors demonstrate that VUN100b treatment partially reverses latency without fully reactivating the virus. Furthermore, they also show that VUN100b treatment drives recognition and killing of latently infected monocytes by autologous CMV-specific T cells from HCMV-seropositive individuals. Based on these observations, authors propose that VUN100b can be potentially used as a therapy to clear the HCMV latent reservoir of transplant patients. Overall this is an interesting preliminary observation which is strongly supported by the data presented in this manuscript. However, lack of any data on the assessment of an in vivo efficacy makes it very difficult to evaluate potential use of VUN100b. There are few previously published reports on latent CMV infection in humanized murine model. I was wondering if authors have considered using one of these models to assess the therapeutic potential of VUN100b. I hope authors would agree that there are many therapeutic drugs which show high efficacy in in vitro but often fail in a pre-clinical or clinical setting.

We thank the reviewer for reading our manuscript and finding our results interesting. We agree with the reviewer that further preclinical and clinical (in vivo) testing would give even more conclusive results regarding the therapeutic potential of the nanobody. Although there is a humanized murine model, as suggested by the reviewer, we would like to point out that this model has shown different results regarding the role of US28 in HCMV latency and reactivation (Crawford et al., mBio 2019) as opposed to those from our group and other groups (Humby et al., J. Virol. 2015; Krishna et al., mBio 2018; Krishna et al., PNAS 2019; Zhu et al., Nat. Microbiol. 2018). We agree that it would be interesting to see potency of the US28 nanobodies in this humanized mouse model, giving potentially more information regarding the specific role of US28 in latency and reactivation. However, we are unsure whether these humanized models would give conclusive results in respect to our experimental setup. Nevertheless, we are currently looking into multiple possibilities to test these molecules in more (pre)clinical settings, to further demonstrate their therapeutic potential.

Reviewer #3 (Remarks to the Author):

In this paper, the authors describe the effect of a bivalent nanobody (VUN100b) that acts as an inverse agonist on its US28 target, a HCMV membrane receptor on latently infected cells. The VUN100b binding to US28 also prevents or disrupts the binding of this viral receptor to host ligands such as CX3CL1. Upon antigen interaction with VUN100b, US28 reactivates some, but not all, genes of HCMV. Interestingly no (or very low numbers) of infective HCMV particles are produced. Surprisingly, treatment of latently infected monocytes with this bivalent nanobody in presence of CTLs from HCMV seropositive persons will severely reduce the HCMV reservoir cells. Thus, this VUN100b construct could be developed as a therapeutic tool to remove HCMV latently infected cells in transplant patients.

It is an interesting and unexpected observation, the experiments are well designed and provide convincing evidence to support the conclusions.

My expertise is directed more towards antibody engineering, rather than virology. An antibody engineer will not find much details in this paper as the reader is directed to a previous paper (De Groof et al, Mol Pharm, 2019) where this nanobody has been characterised in more detail. Here this nanobody was made bivalent to obtain an increased binding due to avidity effects. Of note, it would be better to name this bivalent nanobody VUN100bv (instead of VUN100b), this way it is more clear and a difference can be made between bivalent (bv) bispecific (bs) and biparatopic (bp) as such constructs might be preferred in the future. Also, since this bivalent construct is more effective than the monovalent nanobody, it might be a good idea to communicate the expression yield of the bivalent construct (relative to monomer). Bivalent nanobody constructs are expected to have a lower yield, while the same effect can be observed by adding slightly larger amounts of the monovalent nanobody on the cells. It would be very interesting to see the crystal structure of the (monomeric) nanobody in complex with SU28. However, it is conceivable that the crystallisation of this challenging protein (complex) will require an enormous effort, and might not be feasible in the end. Nevertheless, such crystal structure will provide the fine details of the exact epitope, and might be the start to understand the intriguing inverse agonist behaviour of this nanobody.

We thank the reviewer for the positive feedback on our manuscript and finding our results interesting and the experiments well designed. We agree with the comment that VUN100bv would be a more appropriate abbreviation and we have changed this in the manuscript.

The reviewer is correct in the assessment that yields between the monovalent and bivalent format of the VUN100 nanobody differ. We see that similar amounts of starting cultures of E Coli result in a 5-fold higher expression yield of monovalent nanobody compared to the bivalent format. We have added this notion in the manuscript.

To address the question regarding the effect of higher concentrations of the monovalent VUN100 nanobody on US28 activity, we have performed an additional control experiment using 1 μ M of monovalent VUN100. This concentration corresponds to 400-fold its K_d value. Also at this high concentration, monovalent VUN100 did not show any effect on US28 activity. This data has been added as Supplementary figure 2.

Finally, we agree with the reviewer that a crystal structure of the nanobody in complex with US28 would be very interesting. These results would give us more information regarding the epitope and could explain the functional effects of the nanobody on receptor signaling. However, obtaining crystal structures of GPCRs is highly challenging and is dependent on the stabilization of the receptor to reduce conformational heterogeneity (Manglik et al., Annu Rev Pharmacol Toxicol. 2017). So far, attempts to obtain such structures with VUN100 has proven to be challenging as well.

REVIEWER COMMENTS

Reviewer #1 (Remarks to the Author):

In this revised manuscript by De Groof, et al, the authors have made improvements based on critiques from reviewers. The authors have improved their methods section, added a few control experiments, and made additional adjustments requested by the reviewers. While one reviewer suggested the use of a humanized mouse model, I support the authors' decision in choosing not to include this experiment in the current manuscript. While the manuscript is somewhat improved, it still suffers from a lack of organization and highly qualitative assays to evaluate stages of infection for which there are well-documented and well-accepted quantitative assays. Specific, remaining outstanding points are provided for the authors' consideration.

Major:

1. As indicated in the initial review, the experimentation remain highly qualitative. This is really a shame, because if the authors employed the very well-established quantitative assays, with the proper controls, enthusiasm would really be bolstered.
2. As mentioned in the first round of reviews, the authors say throughout in several places that they quantified full reactivation and subsequent virus production. They did not. They measured IE protein expression by immunofluorescence. This is not a measure of full reactivation or virus production. This is the ability for IE protein to be expressed. As mentioned on the first round of review, to measure reactivation, which is not assessed anywhere in this manuscript, the authors must do the appropriate assay, such as quantifying plaques. Again, as mentioned in the first reviews, IE protein production does not always lead to a successful reactivation and/or lytic infection – CMV often leads to abortive infections and/or spontaneous reactivation (IE positive, but no plaques develop). Thus, without quantification of plaques, the authors' claims that the virus only partially reactivates are completely unsupported. While they may argue they fail to see viral late genes made, the assessment of a single gene of each transcriptional class is not enough data to support this conclusion, and I would argue transcriptional data alone does not allow one to decipher how far an infection has gotten in the reactivation stage (even further supported by the fact that a "latent transcriptional profile" has been challenged in recent years). Finally, as the authors have now provided additional methods on this round of revision, it is even more clear this is the less-than-ideal assay, as they have evaluated this in a 96-well format, which means they are looking at an incredibly small number of cells. The authors also fail to indicate how many cells are plated in this very small well, making it difficult to understand what their cell numbers counted by IE2yfp or UL32gfp actually mean. Even if the authors were adamant about this methodology (which I strongly discourage), it would be far more accurate to plate more cells in a larger well format, as accuracy decreases with well size in such qualitative assays. This may indeed lead to my trepidations on the UL32gfp data - I am not convinced the UL32-gfp viral infections are actually latent to begin with. This again could be rectified if the authors performed quantitative rather than qualitative assays. I remain skeptical of the high number of gfp-positive cells in these cultures – UL32 is a late gene, thus is completely unclear why so much late protein is made in comparison to IE2 protein in the cells infected with the IE2yfp virus. Again, quantitative assays that actually measure and confirm latency vs. reactivation would rectify this, but those experiments are not included.
3. The data in Fig 3 is not comparable. The authors use a WT virus in one backbone (HCMV IE2-eYFP) and mutant (delUS28) virus in a completely different background (Titan). These are not comparable. The mutant must be used in conjunction with the matched parental virus backbone - the mutant in this experiment isn't even in the same HCMV strain! This data is completely uninterpretable, as the experiment was simply not performed correctly.
4. The authors should take the time to reorganize this manuscript. Some of the added data that was added to the end should really be combined with the complementary data presented earlier. Furthermore, it is unclear why the authors present control data separate from the actual experiment. This is confusing. Supp Fig 7 needs to be included/combined with the data in Fig 5a – the experiment, including the control, must be performed together at the same time and thus plotted together. Hence, a proper control. The same goes for Supp Fig 9 – this needs to be shown WITH the experimental dataset – it must be performed side-by-side and plotted together. Such data is irrelevant and confusing when presented alone.
5. While the authors have added rationale for assessing NFAT activity, as well as the experiment

shown in Fig 1C, it is important for the authors to note that NFAT activity has not been confirmed in the context of infection (per the reference they provided and a little digging in the literature), nor has it been confirmed herein in the context of the nanobody experiments. First, this experiment, like the referenced work, was performed in HEK293Ts – in this current manuscript, this experiment really should have been performed in THP-1 cells (like the data in Fig 1E was). US28 signaling is very cell type specific, and HEK293Ts are probably not the most relevant cell type for latency and reactivation, so it is unclear why these were chosen. I get that it is a proof of principle experiment, but cell type matters, especially when the data in Fig 1E,F is not very convincing. Further, this experiment is confusing – the authors apparently looked at “US28 activity” (per their Fig 1C y-axis label) by transient transfection of HEK293Ts with an NFAT luciferase reporter construct and the respective US28 construct noted on the x-axis. The ELISA is for HA – the tag on the US28 protein, whether it be wildtype or mutant. Where is the NFAT activity data in this context? If they are measuring HA by ELISA, that is a simple measure of US28 expression, not activity (though arguable the better way to measure US28 expression is by western for HA). And if this is correct, this brings up even more questions, because one would think the expression of US28 would be similar across the board (especially since the cells were permeabilized, so this isn't surface expression). I fail to see how this measures the effect of nanobodies on US28-mediated NFAT expression if the authors do not measure NFAT activity (luciferase), as indicated in the text and legend, in this system. Perhaps I have misunderstood this – and if so, then this needs to be made far clearer than it is currently.

6. Figure 4 controls remain missing. Non-treated control samples, non-cocultured samples in b/c. These are important to data interpretation. The same can be said for Fig 5c-e.

Minor:

1. The authors have added information about replicates, as requested on initial review. It appears many experiments only have two biological replicates – at a minimum, three biological replicates should be performed. The authors have elected to show representative data and do not combine the findings, which they state is due to varying baselines. While that may be, concluding that across the board when the experiments have only been done two times seems premature.
2. Supp Fig 6 – DNA genome maintenance (panel B) needs to be measured prior to co-culture so one can evaluate what the “after” co-culture data means. This information simply cannot be garnered from IE positive cell number (shown in A).

Reviewer #3 (Remarks to the Author):

In this revised version, the authors responded in an adequate manner to the comments and criticisms raised by the reviewers.

I have no further comments except that in future papers the authors should not refer to 'irrelevant' nanobodies. If the nanobodies are irrelevant then they should not have been used; What they mean is perhaps 'non-targeting' nanobodies used as a control. Such control molecules are indeed relevant.

Reviewer #1 (Remarks to the Author):

In this revised manuscript by De Groof, et al, the authors have made improvements based on critiques from reviewers. The authors have improved their methods section, added a few control experiments, and made additional adjustments requested by the reviewers. While one reviewer suggested the use of a humanized mouse model, I support the authors' decision in choosing not to include this experiment in the current manuscript. While the manuscript is somewhat improved, it still suffers from a lack of organization and highly qualitative assays to evaluate stages of infection for which there are well-documented and well-accepted quantitative assays. Specific, remaining outstanding points are provided for the authors' consideration.

We thank the reviewer for his/her comments and recognition that our changes improved the paper. We address the remaining criticisms below.

Major:

1. As indicated in the initial review, the experimentation remain highly qualitative. This is really a shame, because if the authors employed the very well-established quantitative assays, with the proper controls, enthusiasm would really be bolstered.

We agree with the reviewer that, many of the analyses regarding activation of IE expression and subsequent T cell killing of these IE-expressing cells rely on manual counting of IE-eYFP positive cells, but disagree with the statement that this method is highly qualitative rather than quantitative. Analysis of infectious virus, be it by TCID₅₀, focus-formation or infectious centre assays (included in our manuscript), and plaque assay all require the manual assessment and counting of viral protein production or cell lysis. Nevertheless, as manual counting (of plaques or IE-positive infectious centres) can be subject to operator bias and as requested by the reviewer, we have added new experiments that demonstrates that VUN100bv drives the phenomena presented in the manuscript using unbiased and more quantitative methods.

In general, the assessment of cytomegalovirus latency and reactivation requires sensitive methods to analyse because the levels of virus are low. We are also limited by amount of starting material. Hence, there are ethical considerations to take into account when obtaining blood donations from a seropositive individual. We wish to reassure the reviewer that we take as much material as is ethically justifiable by our ethics permit and use 90-100% of the material in each experiment.

We would also like to indicate that the used method (counting of IE-positive cells) has been used by our lab and others and has been peer-reviewed and accepted in multiple journals, including this journal (Krishna et al., Nature Communications, 2017) (Krishna et al., Scientific reports, 2016) (Krishna et al., mBio, 2017) (Kew et al., Scientific Reports, 2017). As the initial seeded amount of CD14+ monocytes does not change during the period of the assay, we have chosen not to show our results as percentage of infected cells taking into account the total amount of cells.

To further address the reviewers' comments we have performed two type of additional experiments. Firstly, we include in this letter data from an experiment similar to that presented in Figure 2 in the main manuscript, but this time using the Thermo Scientific ArrayScan system. This system identifies cell objects based on Hoescht (nuclei stain) signal and reads additional fluorescent signal from other channels (such as that from IE2-eYFP), applying a threshold across all wells for positive cells; that is to say it is an automated plate reader for immunostained cells. These results (Figure A, presented below) recapitulate the counting that we performed manually. We chose not to use this machine for all of our experiments principally because the nuclei stain is ultimately toxic to the cells, preventing the analysis of IE protein expression at later time points in the same wells.

Figure A. CD14+ monocytes were isolated, infected with HCMV IE2-eYFP and treated with an irrelevant nanobody, VUN100 or VUN100bv for six days. Six days post infection, cells were fixed and analysed for immediate early (IE)-expression using the Thermo Fisher ArrayScan system. Total numbers of cells were visualized by Hoechst staining.

Secondly, we have included a set of new analyses based on the analysis of CMV gDNA by PCR (new Figure 5). We determined the HCMV gDNA copy numbers of CD14+ monocytes infected with HCMV and untreated or treated with an irrelevant nanobody, VUN100bv or PMA for 6 days using a 48-wells format as suggested by the reviewer. Using this assay, we were able to show that there is no increase in HCMV gDNA levels upon VUN100bv treatment in contrast to PMA treatment. These experiments validate our previous assays including the fibroblast co-culturing experiments (Figure 2C, 2E) that the VUN100bv nanobody treatment does not result in full reactivation. Moreover, as requested by the reviewer in minor comment 2, we have also quantified HCMV gDNA copy numbers before and after T cell and fibroblast co-culturing to validate our previous results showing that VUN100bv treatment results in the recognition and killing of HCMV-infected CD14+ monocytes. Overall, these unbiased PCR-based analyses fully support results previously obtained by counting IE positive cells (both transient IE reactivation in monocytes and reactivation of infection virus after co-culture with fibroblasts), and show that reactivation of HCMV is ablated when latently infected cells are cocultured with the US28 nanobodies (VUN100bv) and T cells

2. As mentioned in the first round of reviews, the authors say throughout in several places that they quantified full reactivation and subsequent virus production. They did not. They measured IE protein expression by immunofluorescence. This is not a measure of full reactivation or virus production. This is the ability for IE protein to be expressed. As mentioned on the first round of review, to measure reactivation, which is not assessed anywhere in this manuscript, the authors must do the appropriate assay, such as quantifying plaques. Again, as mentioned in the first reviews, IE protein production does not always lead to a successful reactivation and/or lytic infection – CMV often leads to abortive infections and/or spontaneous reactivation (IE positive, but no plaques develop). Thus, without quantification of plaques, the authors' claims that the virus only partially reactivates are completely unsupported.

We would like to apologise to the reviewer that our manuscript was unclear in the wording of how we assessed full reactivation and subsequent virus production (eg Figure 2C). When we have measured virus reactivation, we have done so using the IE focus forming assay in co-cultured fibroblasts, used by many groups to quantify infectious cytomegalovirus production. Importantly, these IE-positive foci result from de novo infection (from reactivating virus) of fibroblasts in the co-cultures. To clarify this further, we have added a figure below (Figure B) explaining the principle of the assay as well as a microscopy picture of the IE focus formation that we are quantifying (Figure C). Specifically, we add fibroblasts to the monocyte cultures and count when IE foci have formed; we do not count single cells as, as the reviewer correctly points out, this does not necessarily indicate a fully reactivated or lytic cell. Foci, that is the infection of multiple cells, can only derive from production and transfer of infectious virus and thus measures full viral reactivation. We have rephrased this in the manuscript and hope we are now clear on assessment of IE reactivation and full reactivation of virus (albeit based on IE-positive infection foci in the fibroblast co-cultures).

[Redacted]

We agree that transcriptional data alone is not sufficient to determine whether an infection is latent or lytic and have thus interpreted these data carefully and always in the context of other methods of analysis (IE focus formation during fibroblast coculture). That said, as requested by the reviewer, we have now measured CMV genome maintenance in the VUN100bv-treated monocytes compared to untreated and irrelevant nanobody-(now termed the non-targeting antibody, as requested by reviewer 3, see below) treated monocyte. This demonstrates that CMV genome replication is not a consequence of VUN100bv treatment.

Finally, as the authors have now provided additional methods on this round of revision, it is even more clear this is the less-than-ideal assay, as they have evaluated this in a 96-well format, which means they are looking at an incredibly small number of cells. The authors also fail to indicate how many cells are plated in this very small well, making it difficult to understand what their cell numbers counted by IE2yfp or UL32gfp actually mean. Even if the authors were adamant about this methodology (which I strongly discourage), it would be far more accurate to plate more cells in a larger well format, as accuracy decreases with well size in such qualitative assays. This may indeed lead to my trepidations on the UL32gfp data - I am not convinced the UL32-gfp viral infections are actually latent to begin with. This again could be rectified if the authors performed quantitative rather than qualitative assays. I remain skeptical of the high number of gfp-positive cells in these cultures – UL32 is a late gene, thus is completely unclear why so much late protein is made in comparison to IE2 protein in the cells infected with the IE2yfp virus. Again, quantitative assays that actually measure and confirm latency vs. reactivation would rectify this, but those experiments are not included.

We agree that the UL32-GFP fluorescence data were difficult to interpret and thus have removed these data from the manuscript. Regarding the experimental setup, we use at least 3 wells (technical replicates) containing 1×10^5 CD14+ monocytes per condition in our assay (using a 96 wells format) (experiments of Figures 2 and 4). We have added this in our material and methods section. We would like to stress that this method has been peer-reviewed in multiple journals and has been used by our lab (Krishna et al., Nature Communications, 2017) (Krishna et al., Scientific reports, 2016) (Poole et al., Cell reports, 2018).

In response to the requests of the reviewer, we have performed additional experiments in which we determined the HCMV gDNA copy numbers of CD14+ monocytes infected with the US2-6 intact UL32-GFP HCMV strain that were left untreated or were treated with either a non-targeting nanobody, VUN100bv or PMA for 6 days and this time using a larger 48-wells format. Using this assay, we were able to show that there is no increase in HCMV gDNA levels upon VUN100bv treatment in contrast to PMA treatment. These experiments validate our previous assays including the fibroblast co-culturing experiments that the VUN100bv nanobody treatment does not result in full reactivation. Moreover, we have also determined the HCMV gDNA copy numbers after T cell co-culturing to validate our previous results. This showed that VUN100bv treatment results in the recognition and killing of HCMV-infected CD14+ monocytes.

3. The data in Fig 3 is not comparable. The authors use a WT virus in one backbone (HCMV IE2-eYFP) and mutant (delUS28) virus in a completely different background (Titan). These are not comparable. The mutant must be used in conjunction with the matched parental virus backbone - the mutant in this experiment isn't even in the same HCMV strain! This data is completely uninterpretable, as the experiment was simply not performed correctly.

We agree with the reviewers' comment. The US28-mutant was used as an extra control but is not necessary in this assay as the PMA treated cells show similar results. We have removed this control from the figure.

4. The authors should take the time to reorganize this manuscript. Some of the added data that was added to the end should really be combined with the complementary data presented earlier. Furthermore, it is unclear why the authors present control data separate from the actual experiment. This is confusing. Supp Fig 7 needs to be included/combined with the data in Fig 5a – the experiment, including the control, must be performed together at the same time and thus plotted together. Hence, a proper control. The same goes for Supp Fig 9 – this needs to be shown WITH the experimental dataset – it must be performed side-by-side and plotted together. Such data is irrelevant and confusing when presented alone.

We have restructured the manuscript according to the reviewer's comment. We want to clarify that the controls were already taken along in the same actual experiment as shown in the main manuscript. In the previous version of the manuscript, we had left these out for the purpose of focusing more on our findings regarding the US28 nanobodies. As requested by the reviewer, we have now included the controls in the main figures.

5. While the authors have added rationale for assessing NFAT activity, as well as the experiment shown in Fig 1C, it is important for the authors to note that NFAT activity has not been confirmed in the context of infection (per the reference they provided and a little digging in the literature), nor has it been confirmed herein in the context of the nanobody experiments. First, this experiment, like the referenced work, was performed in HEK293Ts – in this current manuscript, this experiment really should have been performed in THP-1 cells (like the data in Fig 1E was). US28 signaling is very cell type specific, and HEK293Ts are probably not the most relevant cell type for latency and reactivation, so it is unclear why these were chosen. I get that it is a proof of principle experiment, but cell type matters, especially when the data in Fig 1E,F is not very convincing. Further, this experiment is confusing – the authors apparently looked at “US28 activity” (per their Fig 1C y-axis label) by transient transfection of HEK293Ts with an NFAT luciferase reporter construct and the respective US28 construct noted on the x-axis. The ELISA is for HA – the tag on the US28 protein, whether it be wildtype or mutant. Where is the NFAT activity data in this context? If they are measuring HA by ELISA, that is a simple measure of US28 expression, not activity (though arguable the better way to measure US28 expression is by western for HA). And if this is correct, this brings up even more questions, because one would think the expression of US28 would be similar across the board (especially since the cells were permeabilized, so this isn't surface expression). I fail to see how this measures the effect of nanobodies on US28-mediated NFAT expression if the authors do not measure NFAT activity (luciferase), as indicated in the text and legend, in this system. Perhaps I have misunderstood this – and if so, then this needs to be made far clearer than it is currently.

We fully agree with the reviewer that US28-mediated NFAT signaling has not been shown in a (latent) viral setting and we also do not state this in the manuscript. However, these transfected HEK293T cells are an ideal and established model system for determining the pharmacological properties of the nanobodies on constitutive and ligand-mediated US28 signaling. Our main message here, in response to the reviewers comments from the first revision round, was to show the effects of the bivalent nanobody are affecting the constitutive US28 signaling.

To clarify our experimental setup, we co-transfected HEK293T cells with both a vector expressing HA-tagged US28 and a vector containing the luciferase gene under the control of a NFAT-promotor. The constitutive activity of US28 results in an elevated basal level of NFAT-mediated expression of luciferase. This elevated levels of expressed luciferase was quantified by lysing the cells, adding luciferine and measuring luciferase-mediated luminescence. The experiments on different US28 mutants involved separate transfections. Therefore, to ensure that the differences in NFAT activity were solely due to differences in US28 activity, conditions with comparable receptor expression levels were compared, as confirmed by anti-HA ELISA. We have stated this more clearly in the manuscript. Since quantification of GPCRs by western blotting is known to be troublesome due to their aggregation in sample buffer and SDS-PAGE, we have chosen to detect US28 expression via ELISA. The ELISA assay is a well-established method for determining relative receptor expression levels.

Finally, we have performed additional western blot experiments to further show the effect of the VUN100bv on US28-mediated signaling in the more relevant cell line THP-1. We have used the similar experimental set up used in figure 1E but looked at total IFI16 protein levels. We have shown recently that these proteins are downregulated by US28 to support MIEP suppression (Elder et al., 2020, mBio). These additional data show similar results as Figure 1E although we noticed a partial (upon VUN100 treatment) or full restoration (upon VUN100bv treatment) of IFI16 levels. This observation is in line with the small but significant effect of VUN100 treatment upon IE-expression in a viral setting. This is also included in the discussion section.

6. Figure 4 controls remain missing. Non-treated control samples, no n-cocultured samples in b/c. These are important to data interpretation. The same can be said for Fig 5c-e.

We understand that the reviewer prefers to see all these conditions in every experiment presented (in particular a request to show untreated cell controls), but we wish to remind the reviewer of the ethical limitations on our starting material. Furthermore, as stated in our previous rebuttal, the irrelevant nanobody (now called the non-targeting

antibody throughout the new manuscript as requested by reviewer 3, see below) is the most relevant control for these experiments. This reasoning is shared by the other reviewers. The irrelevant nanobody is purified in the same manner as VUN100bv, and, therefore, is the crucial control for any e.g. contaminating factors in the nanobody production and purification process. As mentioned in the previous rebuttal, a small non-specific effect was seen for the irrelevant nanobody. For that reason, comparing the results between the bivalent VUN100 and the irrelevant nanobody is more correct. That said, in our new quantitative analyses requested by the reviewer in larger well format (new figure 5), we have also included untreated cells that confirm our previous results.

Minor:

1. The authors have added information about replicates, as requested on initial review. It appears many experiments only have two biological replicates – at a minimum, three biological replicates should be performed. The authors have elected to show representative data and do not combine the findings, which they state is due to varying baselines. While that may be, concluding that across the board when the experiments have only been done two times seems premature.

We have performed additional repeats for some assays but we feel that repeating all assays again would be unethical in accordance with our ethical approval from Cambridgeshire 2 Research Ethics Committee (REC reference 97/092). Moreover, we have three biological repeats for the additional control experiments (in 48-wells format) that validate our previous results where we have two biological repeats. We hope the reviewer is satisfied by the fact that we have used multiple donors in independent experiments and that the final three figures, representing 7 independent experiments, show the same phenomena with minor modifications to the experimental protocol (e.g larger well format, as requested by the reviewer) or analysis method (e.g. DNA PCR, also as requested by the reviewer). Overall, we show multiple times similar functional effects of VUN100bv treatment.

2. Supp Fig 6 – DNA genome maintenance (panel B) needs to be measured prior to co-culture so one can evaluate what the “after” co-culture data means. This information simply cannot be garnered from IE positive cell number (shown in A).

As requested by the reviewer, we have measured CMV DNA genome maintenance in this experiment prior to co-culture and these data support our previous conclusions that VUN100bv, specifically, drives T-cell killing and a reduction in CMV reactivation in monocytes (new figure 5).

Reviewer #3 (Remarks to the Author):

In this revised version, the authors responded in an adequate manner to the comments and critics raised by the reviewers.

I have no further comments except that in future papers the authors should not refer to 'irrelevant' nanobodies. If the nanobodies are irrelevant then they should not have been used; What they mean is perhaps 'non-targeting' nanobodies used as a control. Such control molecules are indeed relevant.

We agree with the reviewer that these “irrelevant” nanobodies are indeed the most relevant control and so have changed “irrelevant” to non-targeting nanobodies throughout the manuscript, as requested.

Reviewers' comments:

Reviewer #1 (Remarks to the Author):

This is a revised manuscript from DeGroof et al. The authors have been somewhat responsive to previous critiques, and the clarification on their IE focus formation assay was quite helpful. However, some concerns remain unresolved. Suggestions are provided below:

Major:

1. While PMA controls were added in some places, it appears that the authors added them to data already acquired (e.g. Fig 2B and Fig 4A and 4B (former Fig 4C)), rather than repeating the experiment with proper controls included with the experimental samples. Doing it separately completely defeats the purpose. The authors have justified not performing all controls due to ethical concerns. As raised in previous rounds of review, the authors do not provide details on cell numbers. Regardless, using their references cited in the methods, one can calculate roughly that it is likely they isolate $7 \times 10^6 - 1 \times 10^8$ CD14+ cells from each apheresis cone. So, for experiments that use 5 conditions, each sample in triplicate, using 7×10^6 as the starting number, one would have 4.7×10^5 cells per well. Since most of the experiments seem to be performed in 96 wells (genomic DNA analysis being the exception), this seems more than feasible, and within ethical limitations.
2. The PMA controls for most experiments remain improperly done. As noted in previous reviews, the proper control is to allow the cells to go latent, and then treat with PMA. Pre-treatment tells you absolutely nothing if you are looking to control a reactivation phenotype. As suggested previously, the proper control is latent infection, then PMA treatment. This is essential for interpreting the data.
3. The genomic maintenance experiment is difficult to evaluate. The authors need to include a step to remove virus that bound cells, that did not enter (e.g. citrate wash, trypsin treatment). This is a critical step necessary for such an experiment.
4. Figure 1: As noted in previous reviews, the presentation of the NFAT data is confusing. The authors are doing a luciferase assay. The y-axis on Fig 1C, for example, should reflect that, and not read "US28 activity", which is not being measured. As previously mentioned, NFAT has not been shown in the context of infection to be regulated by US28 (by literature search), and it is unclear why a better output was not chosen (e.g. NFkB, STAT3 – shown to be constitutively regulated by US28 during infection, according to the literature). Further, as mentioned in previous reviews, US28 expression should be evaluated by western blot. It is not clear why the authors find such a standard evaluation "troublesome". A quick scan of the literature shows many labs can successfully detect GPCRs by this method. It also appears by looking through the literature that many labs can even detect US28 by this method. This would help, because the data shown in E aren't very robust (arguably, panel F looks fantastic). It is also quite unclear how a blot in which all samples should've been run together for all repeats, was not (Panel E, where the authors state the data represents 3-4 blots of 3-4 independent experiments) – the bar labeled IrrNb seems to have only 3 data points, while all others have 4. Would suggest confirming US28 expression by western, deleting E and retaining F.

Minor:

1. While the authors have changed "irrelevant nanobody" to "non-targeting nanobody" in some places, it is not corrected throughout. Namely, the authors did not make an effort to change this in the figures, which they should. This would be helpful for the reader.
2. Figure 2 – This figure is hard to follow - there is limited information in the legends, and the text is no easier to follow. Panel D is quite confusing because there are various labels, but it seems from the text these are essentially all untreated samples? As presented, it's quite misleading. It's just not clear (this same comment applies to Fig4c). 2d data should be shown the same as 6d data. IE-focus formation at 2d should be included.
3. Fig S5: The figure legend states the following: "6dpi, cells were untreated..., treated with the [different nanobodies]...or PMA. IE-positive nuclei were counted 3dpi." Suggest editing for actual method used (probably a type-o on one of the time points, but it is confusing as is). Also, IE-focus formation should really be shown here, side-by-side with the IE nuclei numbers, as in the main

text.

4. Various places that state that monocytes were differentiated to "induce full reactivation" (e.g. in reference to Fig 4e) should be changed, as "full reactivation" is not measured such instances.
5. It is unclear why differentiation/reactivation treatments (PMA vs. GM-CSF) are not consistent.

Reviewers' comments:

Reviewer #1 (Remarks to the Author):

This is a revised manuscript from DeGroof et al. The authors have been somewhat responsive to previous critiques, and the clarification on their IE focus formation assay was quite helpful. However, some concerns remain unresolved. Suggestions are provided below:

Major:

1. While PMA controls were added in some places, it appears that the authors added them to data already acquired (e.g. Fig 2B and Fig 4A and 4B (former Fig 4C)), rather than repeating the experiment with proper controls included with the experimental samples. Doing it separately completely defeats the purpose. The authors have justified not performing all controls due to ethical concerns. As raised in previous rounds of review, the authors do not provide details on cell numbers. Regardless, using their references cited in the methods, one can calculate roughly that it is likely they isolate $7 \times 10^6 - 1 \times 10^8$ CD14+ cells from each apheresis cone. So, for experiments that use 5 conditions, each sample in triplicate, using 7×10^6 as the starting number, one would have 4.7×10^5 cells per well. Since most of the experiments seem to be performed in 96 wells (genomic DNA analysis being the exception), this seems more than feasible, and within ethical limitations.

Reviewer 1 states here that we have simply added PMA control generated subsequently to data from our original assays. However, we absolutely state, again, that this is not the case. In our previous response letter to the reviewers, we clearly stated as response to question 4 of reviewer 1: "We want to clarify that the controls were already taken along in the same actual experiment as shown in the main manuscript. In the previous version of the manuscript, we had left these out for the purpose of focusing more on our findings regarding the US28 nanobodies. As requested by the reviewer, we have now included the controls in the main figures." We reiterate that the PMA control samples were generated in the same assays (on the same plate, with the same cells, at the same time) and are the proper controls for the data shown - we have not "added them to data already acquired".

We have already commented on the numbers of cells used in our response to question 2 during the previous round of revisions ("Regarding the experimental setup, we use at least 3 wells (technical replicates) containing 1×10^5 CD14+ monocytes per condition in our assay (using a 96 wells format) (experiments of Figures 2 and 4)."). Moreover, as requested by reviewer 1 in their comments, we have upscaled this to a 48-wells format (tripling the amount of cells used).

What is perhaps not clear to the Reviewer is that it is not always the amount of monocytes that limits the size of our experiments but the amount of **autologous** T cells. This is because our experiments require **autologous** T cells/PBMC that must be frozen for a week and subsequently recovered and separated. In order to achieve the E:T ratio stated of 5:1 we usually use 95-100% of recovered T cells or PBMC. Furthermore, we could not use apheresis cones as the source of PBMC for these T cell-based experiments, instead using arm bleeds from known seropositive donors; this is because apheresis cones obtained from the blood transfusion services are of unknown HCMV serostatus. We apologise that this was unclear in the Materials and Methods, but the use of arm bleeds from known seropositive donors also necessarily limits the numbers of cells available to us for these types of assays due to much reduced volume of blood donated.

Reviewer 1 now comments that we could have easily obtained enough cells to redo all our original analyses but bases their calculations on estimates of cell numbers using a 96 well format. And yet this is despite, Reviewer 1, themselves, specifically requested that repeat analyses use a 48 well format. We did exactly as requested by the reviewer and carried out new core analyses assays using a 48 well format. We could not, for reasons detailed above, repeat all our original analyses in every figure in a 48 well format. In essence, in their most recent review, reviewer 1 back-tracks on their previous request (to use a 48 well format for any further analyses) and argues that we could have used a 96 well format to redo all our original analyses. We reiterate, reviewer 1 specifically requested that we redo analyses in a larger well format.

Reviewer 1 remains critical about our experimental set up despite us clearly showing with multiple references that our methods used are widely accepted in the field. We were able to convince him/her finally that we are looking at full viral reactivation although we needed to reiterate our argument during two rounds of revisions. Again, we would like to stress that our methods used are valid and published in high ranking journals including Nature Communications (Krishna et al., Nature Communications, 2017) (Krishna et al., Scientific reports, 2016) (Krishna et al., mBio, 2017) (Kew et al., Scientific Reports, 2017). Moreover, we have used exactly the same methodology in a paper that has recently been published in PNAS where no issues were raised about the methodology (Groves et al, <https://doi.org/10.1073/pnas.2023025118>).

2. The PMA controls for most experiments remain improperly done. As noted in previous reviews, the proper control is to allow the cells to go latent, and then treat with PMA. Pre-treatment tells you absolutely nothing if you are looking to control a reactivation phenotype. As suggested previously, the proper control is latent infection, then PMA treatment. This is essential for interpreting the data.

Indeed, as stated by reviewer 1, pre-treatment does not tell us anything about reactivation. However, pre-treatment results in the inability of establishing latency (a process where US28 is also critical). We show that our nanobodies (and PMA) influence this process. However, we have performed additional experiments (Figure 2D-F; Figure 4C-E; supplementary figure 5) where we do exactly as requested by reviewer 1, as requested during round 1 of the revisions: we let the cells go latent and clearly show that our nanobody and PMA treatment can reactivate the cells resulting in T cell mediated killing of the reactivated cells. These results reiterate our main findings using the pre-treatment of PMA and show that these different set-ups also give the same results. Importantly, this principle seemed to be accepted by reviewer 1 in the previous review and was also accepted by the two other reviewers.

3. The genomic maintenance experiment is difficult to evaluate. The authors need to include a step to remove virus that bound cells, that did not enter (e.g. citrate wash, trypsin treatment). This is a critical step necessary for such an experiment.

A citrate wash or trypsin treatment is a critical step to remove virus bound to cells when studying virus entry and immediate genome uptake, and is used by herpesvirologists over a period of up to 24 hours post infection. (e.g. <https://jvi.asm.org/content/86/24/13745.short>). We harvest the genome, at the earliest, 6 days post infection (by which time the medium had been replaced 2 times) and extra washes were performed prior to DNA harvesting ensuring that all/most virus was removed. This is in line with how other groups than our own in the HCMV latency field conduct their analyses of HCMV genome maintenance, without a citrate wash or trypsin step (e.g. Crawford et al 2019 <https://www.ncbi.nlm.nih.gov/pmc/articles/PMC6703429/> and Collins-McMillen et al 2019 <https://www.ncbi.nlm.nih.gov/pmc/articles/PMC6717278/>). However, the possibility that low amounts of virus might still be non-specifically present on the cell surface would still be controlled for by the fact that this would be the case for all conditions (as all cells were treated the same way) and, hence, this would not differentially influence the assay outcome.

4. Figure 1: As noted in previous reviews, the presentation of the NFAT data is confusing. The authors are doing a luciferase assay. The y-axis on Fig 1C, for example, should reflect that, and not read "US28 activity", which is not being measured. As previously mentioned, NFAT has not been shown in the context of infection to be regulated by US28 (by literature search), and it is unclear why a better output was not chosen (e.g. NFkB, STAT3 – shown to be constitutively regulated by US28 during infection, according to the literature). Further, as mentioned in previous reviews, US28 expression should be evaluated by western blot. It is not clear why the authors find such a standard evaluation "troublesome". A quick scan of the literature shows many labs can successfully detect GPCRs by this method. It also appears by looking through the literature that many labs can even detect US28 by this method. This would help, because the data shown in E aren't very robust (arguably, panel F looks fantastic). It is also quite unclear how a blot in which all samples should've been run together for all repeats, was not (Panel E, where the authors state the data represents 3-4 blots of 3-4 independent experiments) – the bar labeled Irr Nb seems to have only 3 data points, while all others have 4. Would suggest confirming US28 expression by western, deleting E and retaining F.

We feel that it is clear that reviewer 1 has misinterpreted the assay, here. We have tried to clarify this during the previous round of revisions as well. The luciferase assay is an assay to measure US28 activity as US28 activates the accumulation of luciferase (see our explanation in the previous revision). These types of assays are widely used in the pharmacology field (Unal, *Methods in Cell biology*, 2019; <https://doi.org/10.1016/bs.mcb.2018.08.001>) and the same assay looking into US28 activation of NFAT via luciferase assay has been performed by Mclean et al. (<https://doi.org/10.1016/j.virol.2004.04.027>).

Indeed, NFAT mediated signaling has not been shown in THP-1 cells or latency, as we have clearly stated in our previous response as well. We also do not claim this and clearly say that we use the HEK cells to show the **pharmacological** effect of the nanobody on US28 signaling. However, we feel that reviewer 1 here is being quite unfair. During the first round of revisions, reviewer 1 asked us to show that the effect of the nanobody was on constitutive signaling and that to do this we needed to show that NFAT signaling was due to the constitutive activity by the use of additional signaling mutants. As such, we performed this analysis using additional mutants, thereby fully addressing the initial comment. During the 2nd round of revisions, the use of NFAT signaling was suddenly “an inappropriate assay” and we were then asked to show the effect using additional signaling assays. Therefore, as requested, we used the relevant THP-1 cells and analyzed IFI16 expression. The blots of IFI16 protein clearly validate our findings and reviewer 1 even agrees that this looks “fantastic” (panel F) and thus we stand by our overall finding in this figure that our bivalent nanobody inhibits constitutive US28 signaling. During the course of these revisions, we have also gathered additional data showing that nanobody inhibits US28-activated NF- κ B signaling in HEK293T cells - we could change the NFAT assays with these data, if necessary.

We agree with reviewer 1 that the analyses of ERK signaling by Western blot do not show an extremely large effect. This is due to the small window of phosphorylation of ERK, and, likely, the partial inhibition of US28 signaling by our nanobody. However, as stated by the Reviewer, our IFI16 blots validate these findings. If requested, we can remove these data and only show IFI16 data. We want to assure you that all samples are run at the same time, on the same blot and for all repeats. Reviewer 1 is correct that we only have 3 data points for the non-targeting nanobody and 4 data points for all the other conditions. This is due to a technical error during one of the blots whereby the sample of the non-targeting nanobody was lost during the procedure. This is why we have performed a 4th repeat (with newly made samples). Moreover, as reviewer 1 stated, we have put this in the legend by stating: “the data represents 3-4 blots of 3-4 independent experiments”.

Finally, we do not dispute the fact that GPCRs can be detected with western blot. Indeed, it is possible to do this. However, GPCRs are known to aggregate during western blot making **quantification** of receptor expression problematic. Indeed, US28 has been detected using western blot but more often than not show smears or multiple bands (due to aggregation), again, making direct comparison between samples difficult (Wu and Miller, *Virology*, 2016) (Streblow et al, *JBC*, 2003). Using ELISA assays to detect surface and/or total levels of GPCRs is widely accepted as a robust manner to determine receptor expression levels. We have published this method in a chapter in 2009 in “*Methods in Enzymology*” (Massaung et al., [https://doi.org/10.1016/S0076-6879\(09\)05207-0](https://doi.org/10.1016/S0076-6879(09)05207-0)) showing how to do this. Moreover, we have recently published the same techniques for receptor expression determination of US28 and UL33 (another GPCR encoded by HCMV) (De Groof et al., *Mol Pharm.*, 2019, 10.1021/acs.molpharmaceut.9b00360.) (van Senten et al., *JBC*, 2019, 10.1074/jbc.RA119.007796). Finally, a general protocol can be found online to determine cell surface expression of GPCRs by ELISA (<https://www.protocols.io/view/monitoring-cell-surface-expression-of-gpcr-by-elis-zfef3je>). These references clearly support the view that our methods are valid and widely accepted in the pharmacology field.

Minor:

1. While the authors have changed “irrelevant nanobody” to “non-targeting nanobody” in some places, it is not corrected throughout. Namely, the authors did not make an effort to change this in the figures, which they should. This would be helpful for the reader.

We have not changed this because we feel that the abbreviation NT-Nb could be confusing as this is also used as abbreviation of a type of scan in the medical field. As such, we have remained using Irr Nb. However, we can change this if this would be helpful.

2. Figure 2 – This figure is hard to follow - there is limited information in the legends, and the text is no easier to follow. Panel D is quite confusing because there are various labels, but it seems from the text these are essentially all untreated samples? As presented, it's quite misleading. It's just not clear (this same comment applies to Fig4c). 2d data should be shown the same as 6d data. IE-focus formation at 2d should be included.

In Panel D, we do show untreated samples representing the baseline of IE-expressing cells prior to treatment. These data were previously included as supplementary data but this was requested by reviewer 1 to be added in the main manuscript in a previous round of revision. In the previous two reviews, no problems were raised by reviewer 1 about this presentation of the data and we included it in the main figures, as requested by the reviewer. Our original view was that these data were better as supplementary data to keep the main manuscript focussed.

The data we show for 2 dpi (Panel A) are microscopy images, which act as a complement for our numerical data presented at 6 dpi. These data have been present since the first version of our manuscript so it is unclear to us why reviewer 1 now has a problem with this data presentation. That said, we are able to show numerical data for 2 dpi if so desired.

However, to show IE-focus formation at 2 dpi is not possible because the replication cycle of human cytomegalovirus is about 72 hours, and likely longer in differentiated monocytic cells (which we are surprised the reviewer is not aware of). Production of infectious virus simply does not occur within 2 days. Thus, we stand by our focus formation assay after longer periods of time.

3. Fig S5: The figure legend states the following: "6dpi, cells were untreated..., treated with the [different nanobodies]...or PMA. IE-positive nuclei were counted 3dpi." Suggest editing for actual method used (probably a type-o on one of the time points, but it is confusing as is). Also, IE-focus formation should really be shown here, side-by-side with the IE nuclei numbers, as in the main text.

We agree that this is a mistake and this should be "IE-positive nuclei were counted 3 days post treatment". However, we do not understand why we should include IE-focus formation, here. Again, this issue was never raised during the previous round of revisions (yet these data were already in the manuscript). The main point of this figure is to show that the effect of the nanobodies is US28 mediated and due to the specific inhibition of US28. This was specifically requested by reviewer 1 during the first round of revisions (question 4).

4. Various places that state that monocytes were differentiated to "induce full reactivation" (e.g. in reference to Fig 4e) should be changed, as "full reactivation" is not measured such instances.

It is well accepted in the field that the differentiation of monocytes under the conditions we have used induces full virus reactivation in vitro and in vivo. The fact that we did not measure infectious virus production, specifically, does not detract from the fact that we induced full reactivation in these samples by standard differentiation-mediated reactivation conditions and this is as opposed to partial reactivation of IE expression which occurs with nanobody treatments.

5. It is unclear why differentiation/reactivation treatments (PMA vs. GMCSF) are not consistent.

We have used PMA treatment principally to induce reactivation from latency or induction of permissiveness for IE expression. Although not a physiological treatment, PMA treatment is a fast (begins to work immediately) methodology to reactivate latent virus in cells in myeloid cell culture. It has also been used by others to reactivate latently infected cells experimentally (Zhu et al 2018 Nature Microbiology <https://www.nature.com/articles/s41564-018-0131-9>).

On the other hand, cytokine treatment with GM-CSF/IL-4 is a much more physiological way to differentiate monocytes into a defined dendritic cell type. We, therefore, used it in our T cell killing assays as we try to recapitulate the types of cells that would reactivate virus in an in vivo process in our experimental system. This cytokine-based differentiation process, however, requires 7 days and thus does not serve as a useful control in assays where we have looked at viral gene/protein expression within 2-6 days.

Opinion of current comments by Reviewer #1, and rebuttal from the Authors. My comments are highlighted in yellow.

Reviewers' comments:

Reviewer #1 (Remarks to the Author):

This is a revised manuscript from DeGroof et al. The authors have been somewhat responsive to previous critiques, and the clarification on their IE focus formation assay was quite helpful. However, some concerns remain unresolved. Suggestions are provided below:

Major:

1. While PMA controls were added in some places, it appears that the authors added them to data already acquired (e.g. Fig 2B and Fig 4A and 4B (former Fig 4C)), rather than repeating the experiment with proper controls included with the experimental samples. Doing it separately completely defeats the purpose. The authors have justified not performing all controls due to ethical concerns. As raised in previous rounds of review, the authors do not provide details on cell numbers. Regardless, using their references cited in the methods, one can calculate roughly that it is likely they isolate $7 \times 10^6 - 1 \times 10^8$ CD14+ cells from each apheresis cone. So, for experiments that use 5 conditions, each sample in triplicate, using 7×10^6 as the starting number, one would have 4.7×10^5 cells per well. Since most of the experiments seem to be performed in 96 wells (genomic DNA analysis being the exception), this seems more than feasible, and within ethical limitations.

Reviewer 1 states here that we have simply added PMA control generated subsequently to data from our original assays. However, we absolutely state, again, that this is not the case. In our previous response letter to the reviewers, we clearly stated as response to question 4 of reviewer 1: "We want to clarify that the controls were already taken along in the same actual experiment as shown in the main manuscript. In the previous version of the manuscript, we had left these out for the purpose of focusing more on our findings regarding the US28 nanobodies. As requested by the reviewer, we have now included the controls in the main figures." We reiterate that the PMA control samples were generated in the same assays (on the same plate, with the same cells, at the same time) and are the proper controls for the data shown - we have not "added them to data already acquired".

We have already commented on the numbers of cells used in our response to question 2 during the previous round of revisions ("Regarding the experimental setup, we use at least 3 wells (technical replicates) containing 1×10^5 CD14+ monocytes per condition in our assay (using a 96 wells format) (experiments of Figures 2 and 4)."). Moreover, as requested by reviewer 1 in their comments, we have upscaled this to a 48-wells format (tripling the amount of cells used).

What is perhaps not clear to the Reviewer is that it is not always the amount of monocytes that limits the size of our experiments but the amount of **autologous** T cells. This is because our experiments require **autologous** T cells/PBMC that must be frozen for a week and subsequently recovered and separated. In order to achieve the E:T ratio stated of 5:1 we usually use 95-100% of recovered T cells or PBMC. Furthermore, we could not use apheresis cones as the source of PBMC for these T cell-based experiments, instead using arm bleeds from known seropositive donors; this is because apheresis cones obtained from the blood transfusion services are of unknown HCMV serostatus. We apologise that this was unclear in the Materials and Methods, but the use of arm bleeds from known seropositive donors also necessarily limits the numbers of cells available to us for these types of assays due to much reduced volume of blood donated.

Reviewer 1 now comments that we could have easily obtained enough cells to redo all our original analyses but bases their calculations on estimates of cell numbers using a 96 well format. And yet this is despite, Reviewer 1, themselves, specifically requested that repeat analyses use a 48 well format. We did exactly as requested by the reviewer and carried out new core analyses assays using a 48 well format. We could not, for reasons detailed above, repeat all our original analyses in every figure in a 48 well format. In essence, in their most recent review, reviewer 1 back-tracks on their previous request (to use a 48 well format for any further analyses) and argues that we could have used a 96 well format to redo all our original analyses. We reiterate, reviewer 1 specifically requested that we redo analyses in a larger well format.

Reviewer 1 remains critical about our experimental set up despite us clearly showing with multiple references that our methods used are widely accepted in the field. We were able to convince him/her finally that we are looking at full viral reactivation although we needed to reiterate our argument during two rounds of revisions. Again, we would like to stress that our methods used are valid and published in high ranking journals including Nature Communications (Krishna et al., Nature Communications, 2017) (Krishna et al., Scientific reports, 2016) (Krishna et al., mBio, 2017) (Kew et al., Scientific Reports, 2017). Moreover, we have used exactly the same methodology in a paper that has recently been published in PNAS where no issues were raised about the methodology (Groves et al, <https://doi.org/10.1073/pnas.2023025118>).

Opinion:

The authors were requested by Reviewer #1 to include PMA controls. The Authors have added these PMA controls and have clearly stated that these controls were performed in the original assays from which the manuscript data was derived. The Authors strongly deny they have done these controls separately and added these controls to previous data. It is my view that the Author's assertion should be taken on face value and thus feel that the Authors have adequately addressed this point.

There are several other points that support the Author's contention that they have adequately addressed the Reviewer's comments raised under this point:

The justification of why all the controls were not included is adequately addressed by the Authors in an explanation of how they have set-up their cultures. Furthermore, they followed the Reviewer's original suggestion to redo their analyses in a larger well format (ie from 96-well to 48-well format).

In addition, the Authors re-state they are using methods that have been peer-reviewed and accepted by high quality journals. They are using the same methodology as the PNAS paper by a highly respected HCMV research lab (Wills group).

I believe the authors have adequately justified the experimental approach they have used in the current manuscript.

2. The PMA controls for most experiments remain improperly done. As noted in previous reviews, the proper control is to allow the cells to go latent, and then treat with PMA. Pre-treatment tells you absolutely nothing if you are looking to control a reactivation phenotype. As suggested previously, the proper control is latent infection, then PMA treatment. This is essential for interpreting the data.

Indeed, as stated by reviewer 1, pre-treatment does not tell us anything about reactivation. However, pre-treatment results in the inability of establishing latency (a process where US28 is also critical). We show that our nanobodies (and PMA) influence this process. However, we have performed additional experiments (Figure 2D-F; Figure 4C-E; supplementary figure 5) where we do exactly as requested by reviewer 1, as requested during round 1 of the revisions: we let the cells go latent and clearly show that our nanobody and PMA treatment can reactivate the cells resulting in T cell mediated killing of the reactivated cells. These results reiterate our main findings using the pre-treatment of PMA and show that these different set-ups also give the same results. Importantly, this principle seemed to be accepted by reviewer 1 in the previous review and was also accepted by the two other reviewers.

Opinion:

The Authors have included additional experiments (Figure 2D-F and Figure 4C-E) where they have allowed the establishment of a latent infection in the cells prior to treatment with PMA (control) and their nanobody, which results in the reactivation of virus. These additional experiments were based on comment at previous review. In my opinion this point has been adequately addressed by the Authors with the addition of this extra data.

3. The genomic maintenance experiment is difficult to evaluate. The authors need to include a step to remove virus that bound cells, that did not enter (e.g. citrate wash, trypsin treatment). This is a critical step necessary for such an experiment.

A citrate wash or trypsin treatment is a critical step to remove virus bound to cells when studying virus entry and immediate genome uptake, and is used by herpesvirologists over a period of up to 24 hours

post infection. (e.g. <https://jvi.asm.org/content/86/24/13745.short>). We harvest the genome, at the earliest, 6 days post infection (by which time the medium had been replaced 2 times) and extra washes were performed prior to DNA harvesting ensuring that all/most virus was removed. This is in line with how other groups than our own in the HCMV latency field conduct their analyses of HCMV genome maintenance, without a citrate wash or trypsin step (e.g. Crawford et al 2019 <https://www.ncbi.nlm.nih.gov/pmc/articles/PMC6703429/> and Collins-McMillen et al 2019 <https://www.ncbi.nlm.nih.gov/pmc/articles/PMC6717278/>). However, the possibility that low amounts of virus might still be non-specifically present on the cell surface would still be controlled for by the fact that this would be the case for all conditions (as all cells were treated the same way) and, hence, this would not differentially influence the assay outcome.

Opinion:

Both the Reviewer and Authors make the point that a wash/treatment step to strip cell surface bound virus that does not enter the cell is important, as this may confound subsequent analyses of intracellular virus. I agree that this is **usually** a key control to include. The divergence of opinion between the Reviewer and Authors is whether this step is critical for the experimental design used in this manuscript.

The Authors make several points to support their contention that this control/step was not required: (i) their harvest was at least 6 days post-infection and included several washes (ii) others examining latency have not used such a wash and (iii) all cells were treated in the same way and so if there were any remaining surface bound virus, this would have been accounted for and thus not contribute to the conclusions made from this assay. I put greatest emphasis on points (i) and (iii) and feel that whilst it would have been desirable to have included a citrate wash or trypsin treatment, these are reasonable explanations and feel that the conclusions are highly unlikely to have been altered by the inclusion of such a treatment(s).

4. Figure 1: As noted in previous reviews, the presentation of the NFAT data is confusing. The authors are doing a luciferase assay. The y-axis on Fig 1C, for example, should reflect that, and not read "US28 activity", which is not being measured. As previously mentioned, NFAT has not been shown in the context of infection to be regulated by US28 (by literature search), and it is unclear why a better output was not chosen (e.g. NFkB, STAT3 – shown to be constitutively regulated by US28 during infection, according to the literature). Further, as mentioned in previous reviews, US28 expression should be evaluated by western blot. It is not clear why the authors find such a standard evaluation "troublesome". A quick scan of the literature shows many labs can successfully detect GPCRs by this method. It also appears by looking through the literature that many labs can even detect US28 by this method. This would help, because the data shown in E aren't very robust (arguably, panel F looks fantastic). It is also quite unclear how a blot in which all samples should've been run together for all repeats, was not (Panel E, where the authors state the data represents 3-4 blots of 3-4 independent experiments) – the bar labeled Irr Nb seems to have only 3 data points, while all others have 4. Would suggest confirming US28 expression by western, deleting E and retaining F.

We feel that it is clear that reviewer 1 has misinterpreted the assay, here. We have tried to clarify this during the previous round of revisions as well. The luciferase assay is an assay to measure US28 activity as US28 activates the accumulation of luciferase (see our explanation in the previous revision). These types of assays are widely used in the pharmacology field (Unal, Methods in Cell biology, 2019; <https://doi.org/10.1016/bs.mcb.2018.08.001>) and the same assay looking into US28 activation of NFAT via luciferase assay has been performed by Mclean et al. (<https://doi.org/10.1016/j.virol.2004.04.027>).

Indeed, NFAT mediated signaling has not been shown in THP-1 cells or latency, as we have clearly stated in our previous response as well. We also do not claim this and clearly say that we use the HEK cells to show the **pharmacological** effect of the nanobody on US28 signaling. However, we feel that reviewer 1 here is being quite unfair. During the first round of revisions, reviewer 1 asked us to show that the effect of the nanobody was on constitutive signaling and that to do this we needed to show that NFAT signaling was due to the constitutive activity by the use of additional signaling mutants. As such, we performed this analysis using additional mutants, thereby fully addressing the initial comment. During the 2nd round of revisions, the use of NFAT signaling was suddenly "an inappropriate assay" and we were then asked to show the effect using additional signaling assays. Therefore, as requested, we used the relevant THP-1 cells and analyzed IFI16 expression. The blots of IFI16 protein clearly validate our findings and reviewer 1 even agrees that this looks "fantastic" (panel F) and thus we stand by our overall

finding in this figure that our bivalent nanobody inhibits constitutive US28 signaling. During the course of these revisions, we have also gathered additional data showing that nanobody inhibits US28-activated NF- κ B signaling in HEK293T cells - we could change the NFAT assays with these data, if necessary.

We agree with reviewer 1 that the analyses of ERK signaling by Western blot do not show an extremely large effect. This is due to the small window of phosphorylation of ERK, and, likely, the partial inhibition of US28 signaling by our nanobody. However, as stated by the Reviewer, our IFI16 blots validate these findings. If requested, we can remove these data and only show IFI16 data. We want to assure you that all samples are run at the same time, on the same blot and for all repeats. Reviewer 1 is correct that we only have 3 data points for the non-targeting nanobody and 4 data points for all the other conditions. This is due to a technical error during one of the blots whereby the sample of the non-targeting nanobody was lost during the procedure. This is why we have performed a 4th repeat (with newly made samples). Moreover, as reviewer 1 stated, we have put this in the legend by stating: "the data represents 3-4 blots of 3-4 independent experiments".

Finally, we do not dispute the fact that GPCRs can be detected with western blot. Indeed, it is possible to do this. However, GPCRs are known to aggregate during western blot making **quantification** of receptor expression problematic. Indeed, US28 has been detected using western blot but more often than not show smears or multiple bands (due to aggregation), again, making direct comparison between samples difficult (Wu and Miller, *Virology*, 2016) (Streblow et al, *JBC*, 2003). Using ELISA assays to detect surface and/or total levels of GPCRs is widely accepted as a robust manner to determine receptor expression levels. We have published this method in a chapter in 2009 in "Methods in Enzymology" (Massaung et al., [https://doi.org/10.1016/S0076-6879\(09\)05207-0](https://doi.org/10.1016/S0076-6879(09)05207-0)) showing how to do this. Moreover, we have recently published the same techniques for receptor expression determination of US28 and UL33 (another GPCR encoded by HCMV) (De Groof et al., *Mol Pharm.*, 2019, 10.1021/acs.molpharmaceut.9b00360.) (van Senten et al., *JBC*, 2019, 10.1074/jbc.RA119.007796). Finally, a general protocol can be found online to determine cell surface expression of GPCRs by ELISA (<https://www.protocols.io/view/monitoring-cell-surface-expression-of-gpcr-by-elis-zfef3je>). These references clearly support the view that our methods are valid and widely accepted in the pharmacology field.

Opinion:

Whilst I do not have access to the previous review round comments, based on the comments above, my view is that the Authors have provided an acceptable response to the issues raised, including a well-justified response regarding the issues surrounding US28 detection by Western blot and the complexities in using such an approach to quantify levels of expression. A couple of points:

It is not unreasonable for the Authors to change the Y-axis label of Figure 1C to indicate luciferase expression.

The authors have gone on further as suggested by the Reviewer to generate additional data showing that the nanobody inhibits US28 signalling in HEK293T cells. This could be added but is not essential as the IFI16 assessment is clear in Panel F.

Minor:

1. While the authors have changed "irrelevant nanobody" to "non-targeting nanobody" in some places, it is not corrected throughout. Namely, the authors did not make an effort to change this in the figures, which they should. This would be helpful for the reader.

We have not changed this because we feel that the abbreviation NT-Nb could be confusing as this is also used as abbreviation of a type of scan in the medical field. As such, we have remained using Irr Nb. However, we can change this if this would be helpful.

Opinion:

For consistency, the Authors should ensure "non-targeting antibody", or an appropriately defined abbreviation (for labelling figures) is made throughout the manuscript.

2. Figure 2 – This figure is hard to follow - there is limited information in the legends, and the text is no easier to follow. Panel D is quite confusing because there are various labels, but it seems from the text these are essentially all untreated samples? As presented, it's quite misleading. It's just not clear (this same comment applies to Fig4c). 2d data should be shown the same as 6d data. IE-focus formation at 2d should be included.

In Panel D, we do show untreated samples representing the baseline of IE-expressing cells prior to treatment. These data were previously included as supplementary data but this was requested by reviewer 1 to be added in the main manuscript in a previous round of revision. In the previous two reviews, no problems were raised by reviewer 1 about this presentation of the data and we included it in the main figures, as requested by the reviewer. Our original view was that these data were better as supplementary data to keep the main manuscript focussed.

The data we show for 2 dpi (Panel A) are microscopy images, which act as a complement for our numerical data presented at 6 dpi. These data have been present since the first version of our manuscript so it is unclear to us why reviewer 1 now has a problem with this data presentation. That said, we are able to show numerical data for 2 dpi if so desired.

However, to show IE-focus formation at 2 dpi is not possible because the replication cycle of human cytomegalovirus is about 72 hours, and likely longer in differentiated monocytic cells (which we are surprised the reviewer is not aware of). Production of infectious virus simply does not occur within 2 days. Thus, we stand by our focus formation assay after longer periods of time.

Opinion:

It is difficult for me to comment specifically about the previous reviewer rounds, as I don't have access to this information. However, I agree with the Reviewer that Panel D is confusing, as the current labelling gives the impression that the cells received the various treatments, when the all the cells shown in this panel are untreated. My preference would be for Panel D to be moved to Supplementary data (as the Authors had preferred), and the X-axis re-labelled (eg by adding the words something along the lines of "prior to addition of..." for each X-axis label).

With respect to Panel A, I think it would be useful to include a graph showing a summary of the numerical data at 2 days post infection (as there is currently no numerical data in the figure for the 2 day time point). Including this information would enrich the microscopy image data shown in Panel A, and sounds like it will be relatively straight forward to include, based upon the Author's offer to include such data.

With respect to the type of data shown at 2 days post infection, the Authors are correct in their assertion that showing IE-focus formation at 2 days post infection is extremely unlikely to be feasible, given the replication kinetics of HCMV. Thus, I agree that the Author's need only show such data at the day 6 time point.

3. Fig S5: The figure legend states the following: "6dpi, cells were untreated..., treated with the [different nanobodies]...or PMA. IE-positive nuclei were counted 3dpi." Suggest editing for actual method used (probably a type-o on one of the time points, but it is confusing as is). Also, IE-focus formation should really be shown here, side-by-side with the IE nuclei numbers, as in the main text.

We agree that this is a mistake and this should be "IE-positive nuclei were counted 3 days post treatment". However, we do not understand why we should include IE-focus formation, here. Again, this issue was never raised during the previous round of revisions (yet these data were already in the manuscript). The main point of this figure is to show that the effect of the nanobodies is US28 mediated and due to the specific inhibition of US28. This was specifically requested by reviewer 1 during the first round of revisions (question 4).

Opinion:

I don't seem to be able to access the Supplementary figure file, so cant comment specifically on this point.

4. Various places that state that monocytes were differentiated to “induce full reactivation” (e.g. in reference to Fig 4e) should be changed, as “full reactivation” is not measured such instances.

It is well accepted in the field that the differentiation of monocytes under the conditions we have used induces full virus reactivation in vitro and in vivo. The fact that we did not measure infectious virus production, specifically, does not detract from the fact that we induced full reactivation in these samples by standard differentiation-mediated reactivation conditions and this is as opposed to partial reactivation of IE expression which occurs with nanobody treatments.

Opinion:

Whilst full reactivation (as opposed to abortive reactivation) can be induced by treating cells to terminally differentiate, this process is often very inefficient. It would be more appropriate for the Authors to use terminology that states that they treated cells to terminally differentiate so as to provide cellular conditions that are conducive to full reactivation (referencing the link between terminal cellular differentiation and reactivation, as well as referencing this methodology to induce reactivation). Thus, I agree with the Reviewer that the terminology should be changed/clarified.

5. It is unclear why differentiation/reactivation treatments (PMA vs. GM-CSF) are not consistent.

We have used PMA treatment principally to induce reactivation from latency or induction of permissiveness for IE expression. Although not a physiological treatment, PMA treatment is a fast (begins to work immediately) methodology to reactivate latent virus in cells in myeloid cell culture. It has also been used by others to reactivate latently infected cells experimentally (Zhu et al 2018 Nature Microbiology <https://www.nature.com/articles/s41564-018-0131-9>).

On the other hand, cytokine treatment with GM-CSF/IL-4 is a much more physiological way to differentiate monocytes into a defined dendritic cell type. We, therefore, used it in our T cell killing assays as we try to recapitulate the types of cells that would reactivate virus in an in vivo process in our experimental system. This cytokine-based differentiation process, however, requires 7 days and thus does not serve as a useful control in assays where we have looked at viral gene/protein expression within 2-6 days.

Opinion:

The authors provide a rational explanation as to why there are differences with the two treatments, but it would be helpful if they could add a similar explanation to the manuscript text to provide this level of clarification to the reader.

Reviewers' comments:

Reviewer #1 (Remarks to the Author):

This is a revised manuscript from De Groof et al. The authors have been somewhat responsive to previous critiques, and the clarification on their IE focus formation assay was quite helpful. However, some concerns remain unresolved. Suggestions are provided below:

Major:

1. While PMA controls were added in some places, it appears that the authors added them to data already acquired (e.g. Fig 2B and Fig 4A and 4B (former Fig 4C)), rather than repeating the experiment with proper controls included with the experimental samples. Doing it separately completely defeats the purpose. The authors have justified not performing all controls due to ethical concerns. As raised in previous rounds of review, the authors do not provide details on cell numbers. Regardless, using their references cited in the methods, one can calculate roughly that it is likely they isolate $7 \times 10^6 - 1 \times 10^8$ CD14+ cells from each apheresis cone. So, for experiments that use 5 conditions, each sample in triplicate, using 7×10^6 as the starting number, one would have 4.7×10^5 cells per well. Since most of the experiments seem to be performed in 96 wells (genomic DNA analysis being the exception), this seems more than feasible, and within ethical limitations.

Answer: Reviewer 1 states here that we have simply added PMA control generated subsequently to data from our original assays. However, we absolutely state, again, that this is not the case. In our previous response letter to the reviewers, we clearly stated as response to question 4 of reviewer 1: "We want to clarify that the controls were already taken along in the same actual experiment as shown in the main manuscript. In the previous version of the manuscript, we had left these out for the purpose of focusing more on our findings regarding the US28 nanobodies. As requested by the reviewer, we have now included the controls in the main figures." We reiterate that the PMA control samples were generated in the same assays (on the same plate, with the same cells, at the same time) and are the proper controls for the data shown - we have not "added them to data already acquired".

We have already commented on the numbers of cells used in our response to question 2 during the previous round of revisions ("Regarding the experimental setup, we use at least 3 wells (technical replicates) containing 1×10^5 CD14+ monocytes per condition in our assay (using a 96 wells format) (experiments of Figures 2 and 4)."). Moreover, as requested by reviewer 1 in their comments, we have upscaled this to a 48-wells format (tripling the amount of cells used).

What is perhaps not clear to the Reviewer is that it is not always the amount of monocytes that limits the size of our experiments but the amount of **autologous** T cells. This is because our experiments require **autologous** T cells/PBMC that must be frozen for a week and subsequently recovered and separated. In order to achieve the E:T ratio stated of 5:1 we usually use 95-100% of recovered T cells or PBMC. Furthermore, we could not use apheresis cones as the source of PBMC for these T cell-based experiments, instead using arm bleeds from known seropositive donors; this is because apheresis cones obtained from the blood transfusion services are of unknown HCMV serostatus. We apologise that this was unclear in the Materials and Methods, but the use of arm bleeds from known seropositive donors also necessarily limits the numbers of cells available to us for these types of assays due to much reduced volume of blood donated.

Reviewer 1 now comments that we could have easily obtained enough cells to redo all our original analyses but bases their calculations on estimates of cell numbers using a 96 well format. And yet this is despite, Reviewer 1, themselves, specifically requested that repeat analyses use a 48 well format. We did exactly as requested by the reviewer and carried out new core analyses assays using a 48 well format. We could not, for reasons detailed above, repeat all our original analyses in every figure in a 48 well format. In essence, in their most recent review, reviewer 1 back-tracks on their previous request (to use a 48 well format for any further analyses) and argues that we could have used a 96 well format to redo all our original analyses. We reiterate, reviewer 1 specifically requested that we redo analyses in a larger well format.

Reviewer 1 remains critical about our experimental set up despite us clearly showing with multiple references that our methods used are widely accepted in the field. We were able to convince him/her finally that we are looking at full viral reactivation although we needed to reiterate our argument during two rounds of revisions. Again, we would like to stress that our methods used are valid and published in high ranking journals including Nature Communications (Krishna et al., Nature Communications, 2017) (Krishna et al., Scientific reports, 2016) (Krishna et al., mBio, 2017) (Kew et al., Scientific Reports, 2017). Moreover, we have used exactly the same methodology in a paper that has recently been published in PNAS where no issues were raised about the methodology (Groves et al, <https://doi.org/10.1073/pnas.2023025118>).

Opinion:

The authors were requested by Reviewer #1 to include PMA controls. The Authors have added these PMA controls and have clearly stated that these controls were performed in the original assays from which the manuscript data was derived. The Authors strongly deny they have done these controls separately and added these controls to previous data. It is my view that the Author's assertion should be taken on face value and thus feel that the Authors have adequately addressed this point. There are several other points that support the Author's contention that they have adequately addressed the Reviewer's comments raised under this point: The justification of why all the controls were not included is adequately addressed by the Authors in an explanation of how they have set-up their cultures. Furthermore, they followed the Reviewer's original suggestion to redo their analyses in a larger well format (ie from 96-well to 48-well format). In addition, the Authors re-state they are using methods that have been peer-reviewed and accepted by high quality journals. They are using the same methodology as the PNAS paper by a highly respected HCMV research lab (Wills group). I believe the authors have adequately justified the experimental approach they have used in the current manuscript.

We thank Reviewer 4 for his/her assessment. We have also updated the Methods section where we clarify the use of arm bleeds of seropositive donors in the case of our T cell co-culturing experiments.

2. The PMA controls for most experiments remain improperly done. As noted in previous reviews, the proper control is to allow the cells to go latent, and then treat with PMA. Pre-treatment tells you absolutely nothing if you are looking to control a reactivation phenotype. As suggested previously, the proper control is latent infection, then PMA treatment. This is essential for interpreting the data.

Answer: Indeed, as stated by reviewer 1, pre-treatment does not tell us anything about reactivation. However, pre-treatment results in the inability of establishing latency (a process where US28 is also critical). We show that our nanobodies (and PMA) influence this process. However, we have performed additional experiments (Figure 2D-F; Figure 4C-E; supplementary figure 5) where we do exactly as requested by reviewer 1, as requested during round 1 of the revisions: we let the cells go latent and clearly show that our nanobody and PMA treatment can reactivate the cells resulting in T cell mediated killing of the reactivated cells. These results reiterate our main findings using the pre-treatment of PMA and show that these different set-ups also give the same results. Importantly, this principle seemed to be accepted by reviewer 1 in the previous review and was also accepted by the two other reviewers.

Opinion:

The Authors have included additional experiments (Figure 2D-F and Figure 4C-E) where they have allowed the establishment of a latent infection in the cells prior to treatment with PMA (control) and their nanobody, which results in the reactivation of virus. These additional experiments were based on comment at previous review. In my opinion this point has been adequately addressed by the Authors with the addition of this extra data.

We thank Reviewer 4 for his/her assessment.

3. The genomic maintenance experiment is difficult to evaluate. The authors need to include a step to remove virus that bound cells, that did not enter (e.g. citrate wash, trypsin treatment). This is a critical step necessary for such an experiment.

Answer: A citrate wash or trypsin treatment is a critical step to remove virus bound to cells when studying virus entry and immediate genome uptake, and is used by herpesvirologists over a period of up to 24 hours post infection. (e.g. <https://jvi.asm.org/content/86/24/13745.short>). We harvest the genome, at the earliest, 6 days post infection (by which time the medium had been replaced 2 times) and extra washes were performed prior to DNA harvesting ensuring that all/most virus was removed. This is in line with how other groups than our own in the HCMV latency field conduct their analyses of HCMV genome maintenance, without a citrate wash or trypsin step (e.g. Crawford et al 2019 <https://www.ncbi.nlm.nih.gov/pmc/articles/PMC6703429/> and Collins-McMillen et al 2019 <https://www.ncbi.nlm.nih.gov/pmc/articles/PMC6717278/>). However, the possibility that low amounts of virus might still be non-specifically present on the cell surface would still be controlled for by the fact that this would be the case for all conditions (as all cells were treated the same way) and, hence, this would not differentially influence the assay outcome.

Opinion:

Both the Reviewer and Authors make the point that a wash/treatment step to strip cell surface bound virus that does not enter the cell is important, as this may confound subsequent analyses of intracellular virus. I agree that this is **usually** a key control to include. The divergence of opinion between the Reviewer and Authors is whether this step is critical for the experimental design used in this manuscript. The Authors make several points to support their contention that this control/step was not required: (i) their harvest was at least 6 days post-infection and included several washes (ii) others examining latency have not used such a wash and (iii) all cells were treated in the same way and so if there were any remaining surface bound virus, this would have been accounted for and thus not contribute to the conclusions made from this assay. I put greatest emphasis on points (i) and (iii) and feel that whilst it would have been desirable to have included a citrate wash or trypsin treatment, these are reasonable explanations and feel that the conclusions are highly unlikely to have been altered by the inclusion of such a treatment(s).

We thank Reviewer 4 for his/her assessment.

4. Figure 1: As noted in previous reviews, the presentation of the NFAT data is confusing. The authors are doing a luciferase assay. The y-axis on Fig 1C, for example, should reflect that, and not read "US28 activity", which is not being measured. As previously mentioned, NFAT has not been shown in the context of infection to be regulated by US28 (by literature search), and it is unclear why a better output was not chosen (e.g. NFkB, STAT3 – shown to be constitutively regulated by US28 during infection, according to the literature). Further, as mentioned in previous reviews, US28 expression should be evaluated by western blot. It is not clear why the authors find such a standard evaluation "troublesome". A quick scan of the literature shows many labs can successfully detect GPCRs by this method. It also appears by looking through the literature that many labs can even detect US28 by this method. This would help, because the data shown in E aren't very robust (arguably, panel F looks fantastic). It is also quite unclear how a blot in which all samples should've been run together for all repeats, was not (Panel E, where the authors state the data represents 3-4 blots of 3-4 independent experiments) – the bar labeled Irr Nb seems to have only 3 data points, while all others have 4. Would suggest confirming US28 expression by western, deleting E and retaining F.

Answer: We feel that it is clear that reviewer 1 has misinterpreted the assay, here. We have tried to clarify this during the previous round of revisions as well. The luciferase assay is an assay to measure US28 activity as US28 activates the accumulation of luciferase (see our explanation in the previous revision). These types of assays are widely used in the pharmacology field (Unal, *Methods in Cell biology*, 2019; <https://doi.org/10.1016/bs.mcb.2018.08.001>) and the same assay looking into US28 activation of NFAT via luciferase assay has been performed by Mclean et al. (<https://doi.org/10.1016/j.virol.2004.04.027>).

Indeed, NFAT mediated signaling has not been shown in THP-1 cells or latency, as we have clearly stated in our previous response as well. We also do not claim this and clearly say that we use the HEK cells to show the **pharmacological** effect of the nanobody on US28 signaling. However, we feel that reviewer 1 here is being quite unfair. During the first round of revisions, reviewer 1 asked us to show that the effect of the nanobody was on constitutive signaling and that to do this we needed to show that NFAT signaling was due to the constitutive activity by the use of additional signaling mutants. As such, we performed this analysis using additional mutants, thereby fully addressing the initial comment. During

the 2nd round of revisions, the use of NFAT signaling was suddenly “an inappropriate assay” and we were then asked to show the effect using additional signaling assays. Therefore, as requested, we used the relevant THP-1 cells and analyzed IFI16 expression. The blots of IFI16 protein clearly validate our findings and reviewer 1 even agrees that this looks “fantastic” (panel F) and thus we stand by our overall finding in this figure that our bivalent nanobody inhibits constitutive US28 signaling. During the course of these revisions, we have also gathered additional data showing that nanobody inhibits US28-activated NF- κ B signaling in HEK293T cells - we could change the NFAT assays with these data, if necessary.

We agree with reviewer 1 that the analyses of ERK signaling by Western blot do not show an extremely large effect. This is due to the small window of phosphorylation of ERK, and, likely, the partial inhibition of US28 signaling by our nanobody. However, as stated by the Reviewer, our IFI16 blots validate these findings. If requested, we can remove these data and only show IFI16 data. We want to assure you that all samples are run at the same time, on the same blot and for all repeats. Reviewer 1 is correct that we only have 3 data points for the non-targeting nanobody and 4 data points for all the other conditions. This is due to a technical error during one of the blots whereby the sample of the non-targeting nanobody was lost during the procedure. This is why we have performed a 4th repeat (with newly made samples). Moreover, as reviewer 1 stated, we have put this in the legend by stating: “the data represents 3-4 blots of 3-4 independent experiments”.

Finally, we do not dispute the fact that GPCRs can be detected with western blot. Indeed, it is possible to do this. However, GPCRs are known to aggregate during western blot making **quantification** of receptor expression problematic. Indeed, US28 has been detected using western blot but more often than not show smears or multiple bands (due to aggregation), again, making direct comparison between samples difficult (Wu and Miller, *Virology*, 2016) (Streblow et al, *JBC*, 2003). Using ELISA assays to detect surface and/or total levels of GPCRs is widely accepted as a robust manner to determine receptor expression levels. We have published this method in a chapter in 2009 in “Methods in Enzymology” (Massaung et al., [https://doi.org/10.1016/S0076-6879\(09\)05207-0](https://doi.org/10.1016/S0076-6879(09)05207-0)) showing how to do this. Moreover, we have recently published the same techniques for receptor expression determination of US28 and UL33 (another GPCR encoded by HCMV) (De Groof et al., *Mol Pharm.*, 2019, 10.1021/acs.molpharmaceut.9b00360.) (van Senten et al., *JBC*, 2019, 10.1074/jbc.RA119.007796). Finally, a general protocol can be found online to determine cell surface expression of GPCRs by ELISA (<https://www.protocols.io/view/monitoring-cell-surface-expression-of-gpcr-by-elis-zfef3je>). These references clearly support the view that our methods are valid and widely accepted in the pharmacology field.

Opinion:

Whilst I do not have access to the previous review round comments, based on the comments above, my view is that the Authors have provided an acceptable response to the issues raised, including a well justified response regarding the issues surrounding US28 detection by Western blot and the complexities in using such an approach to quantify levels of expression. A couple of points: It is not unreasonable for the Authors to change the Y-axis label of Figure 1C to indicate luciferase expression. The authors have gone on further as suggested by the Reviewer to generate additional data showing that the nanobody inhibits US28 signalling in HEK293T cells. This could be added but is not essential as the IFI16 assessment is clear in Panel F.

We thank Reviewer 4 for the points raised. We have changed the Y-axis of Figure 1C (and Supplementary Figure 2) from “US28 activity” to “NFAT-mediated luciferase expression”. As suggested during the previous “response to reviewers letter”, we have removed the ERK1/2 data (which was a point of discussion) and left the IFI16 western blot data to avoid unnecessary discussions.

Minor:

1. While the authors have changed “irrelevant nanobody” to “non-targeting nanobody” in some places, it is not corrected throughout. Namely, the authors did not make an effort to change this in the figures, which they should. This would be helpful for the reader.

Answer: We have not changed this because we feel that the abbreviation NT-Nb could be confusing as this is also used as abbreviation of a type of scan in the medical field. As such, we have remained using Irr Nb. However, we can change this if this would be helpful.

Opinion:

For consistency, the Authors should ensure “non-targeting antibody”, or an appropriately defined abbreviation (for labelling figures) is made throughout the manuscript.

As suggested, we have changed the abbreviations in the manuscript and figures to NT Nb.

2. Figure 2 – This figure is hard to follow - there is limited information in the legends, and the text is no easier to follow. Panel D is quite confusing because there are various labels, but it seems from the text these are essentially all untreated samples? As presented, it’s quite misleading. It’s just not clear (this same comment applies to Fig4c). 2d data should be shown the same as 6d data. IE-focus formation at 2d should be included.

Answer: In Panel D, we do show untreated samples representing the baseline of IE-expressing cells prior to treatment. These data were previously included as supplementary data but this was requested by reviewer 1 to be added in the main manuscript in a previous round of revision. In the previous two reviews, no problems were raised by reviewer 1 about this presentation of the data and we included it in the main figures, as requested by the reviewer. Our original view was that these data were better as supplementary data to keep the main manuscript focussed.

The data we show for 2 dpi (Panel A) are microscopy images, which act as a complement for our numerical data presented at 6 dpi. These data have been present since the first version of our manuscript so it is unclear to us why reviewer 1 now has a problem with this data presentation. That said, we are able to show numerical data for 2 dpi if so desired.

However, to show IE-focus formation at 2 dpi is not possible because the replication cycle of human cytomegalovirus is about 72 hours, and likely longer in differentiated monocytic cells (which we are surprised the reviewer is not aware of). Production of infectious virus simply does not occur within 2 days. Thus, we stand by our focus formation assay after longer periods of time.

Opinion:

It is difficult for me to comment specifically about the previous reviewer rounds, as I don’t have access to this information. However, I agree with the Reviewer that Panel D is confusing, as the current labelling gives the impression that the cells received the various treatments, when the all the cells shown in this panel are untreated. My preference would be for Panel D to be moved to Supplementary data (as the Authors had preferred), and the X-axis re-labelled (eg by adding the words something along the lines of “prior to addition of...” for each X-axis label.

With respect to Panel A, I think it would be useful to include a graph showing a summary of the numerical data at 2 days post infection (as there is currently no numerical data in the figure for the 2 day time point). Including this information would enrich the microscopy image data shown in Panel A, and sounds like it will be relatively straight forward to include, based upon the Author’s offer to include such data.

With respect to the type of data shown at 2 days post infection, the Authors are correct in their assertion that showing IE-focus formation at 2 days post infection is extremely unlikely to be feasible, given the replication kinetics of HCMV. Thus, I agree that the Author’s need only show such data at the day 6 time point.

We have re-labelled the X-axis of Figure 2D by adding “prior to nanobody/PMA treatment” and moved Figure 2D to the supplemental data again. To be consistent, we have done the same for Figure 4C where similar data is shown prior to nanobody/PMA treatment and T-cell co-culturing (now Supplementary Figure 8). We have also added the numerical data of IE positive cells after 2 dpi and 6 dpi as graphs to Supplementary Figure 5.

3. Fig S5: The figure legend states the following: “6dpi, cells were untreated..., treated with the [different nanobodies]...or PMA. IE-positive nuclei were counted 3dpi.” Suggest editing for actual method used (probably a type-o on one of the time points, but it is confusing as is). Also, IE-focus formation should really be shown here, side-by-side with the IE nuclei numbers, as in the main text.

Answer: We agree that this is a mistake and this should be "IE-positive nuclei were counted 3 days post treatment". However, we do not understand why we should include IE-focus formation, here. Again, this issue was never raised during the previous round of revisions (yet these data were already in the manuscript). The main point of this figure is to show that the effect of the nanobodies is US28 mediated and due to the specific inhibition of US28. This was specifically requested by reviewer 1 during the first round of revisions (question 4).

Opinion:

I don't seem to be able to access the Supplementary figure file, so cant comment specifically on this point.

4. Various places that state that monocytes were differentiated to "induce full reactivation" (e.g. in reference to Fig 4e) should be changed, as "full reactivation" is not measured such instances.

Answer: It is well accepted in the field that the differentiation of monocytes under the conditions we have used induces full virus reactivation in vitro and in vivo. The fact that we did not measure infectious virus production, specifically, does not detract from the fact that we induced full reactivation in these samples by standard differentiation-mediated reactivation conditions and this is as opposed to partial reactivation of IE expression which occurs with nanobody treatments.

Opinion:

Whilst full reactivation (as opposed to abortive reactivation) can be induced by treating cells to terminally differentiate, this process is often very inefficient. It would be more appropriate for the Authors to use terminology that states that they treated cells to terminally differentiate so as to provide cellular conditions that are conducive to full reactivation (referencing the link between terminal cellular differentiation and reactivation, as well as referencing this methodology to induce reactivation). Thus, I agree with the Reviewer that the terminology should be changed/clarified.

We thank the reviewer for this suggestion. We have changed the terminology throughout the manuscript by replacing "full reactivation" by "differentiation to allow viral reactivation".

5. It is unclear why differentiation/reactivation treatments (PMA vs. GM-CSF) are not consistent.

Answer: We have used PMA treatment principally to induce reactivation from latency or induction of permissiveness for IE expression. Although not a physiological treatment, PMA treatment is a fast (begins to work immediately) methodology to reactivate latent virus in cells in myeloid cell culture. It has also been used by others to reactivate latently infected cells experimentally (Zhu et al 2018 Nature Microbiology <https://www.nature.com/articles/s41564-018-0131-9>).

On the other hand, cytokine treatment with GM-CSF/IL-4 is a much more physiological way to differentiate monocytes into a defined dendritic cell type. We, therefore, used it in our T cell killing assays as we try to recapitulate the types of cells that would reactivate virus in an in vivo process in our experimental system. This cytokine-based differentiation process, however, requires 7 days and thus does not serve as a useful control in assays where we have looked at viral gene/protein expression within 2-6 days.

Opinion:

The authors provide a rational explanation as to why there are differences with the two treatments, but it would be helpful if they could add a similar explanation to the manuscript text to provide this level of clarification to the reader.

We have added our explanation in the manuscript to further clarify our different treatment methods.